# Global Safe Sequential Learning via Efficient Knowledge Transfer

**Cen-You Li**  *cen-you.li@campus.tu-berlin.de*
*Technical University of Berlin, Germany*
*Bosch Center for Artificial Intelligence, Germany*

**Olaf Duennbier**  *olaf.duennbier@de.bosch.com*
*Robert Bosch GmbH, Germany*

**Marc Toussaint**  *toussaint@tu-berlin.de*
*Technical University of Berlin, Germany*

**Barbara Rakitsch**[*]  *barbara.rakitsch@de.bosch.com*
*Bosch Center for Artificial Intelligence, Germany*

**Christoph Zimmer**[*]  *christoph.zimmer@de.bosch.com*
*Bosch Center for Artificial Intelligence, Germany*

**Reviewed on OpenReview:** *https://openreview.net/forum?id=PtD2gVmb3J*

## Abstract

Sequential learning methods, such as active learning and Bayesian optimization, aim to select the most informative data for task learning. In many applications, however, data selection is constrained by unknown safety conditions, motivating the development of safe learning approaches. A promising line of safe learning methods uses Gaussian processes to model safety conditions, restricting data selection to areas with high safety confidence. However, these methods are limited to local exploration around an initial seed dataset, as safety confidence centers around observed data points. As a consequence, task exploration is slowed down and safe regions disconnected from the initial seed dataset remain unexplored. In this paper, we propose safe transfer sequential learning to accelerate task learning and to expand the explorable safe region. By leveraging abundant offline data from a related source task, our approach guides exploration in the target task more effectively. We also provide a theoretical analysis to explain why single-task method cannot cope with disconnected regions. Finally, we introduce a computationally efficient approximation of our method that reduces runtime through pre-computations. Our experiments demonstrate that this approach, compared to state-of-the-art methods, learns tasks with lower data consumption and enhances global exploration across multiple disjoint safe regions, while maintaining comparable computational efficiency.

## 1 Introduction

Despite the great success of machine learning, acquiring data remains a significant challenge. One prominent approach is to consider experimental design (Lindley, 1956; Chaloner & Verdinelli, 1995; Brochu et al., 2010). In particular, active learning (AL) (Krause et al., 2008; Kumar & Gupta, 2020) and Bayesian optimization (BO) (Brochu et al., 2010; Snoek et al., 2012) resort to a sequential data selection process in which the most informative data points are incrementally added to the dataset. The methods begin with a small dataset, iteratively compute an acquisition function to prioritize data points for querying, select new data based

---

[*]Equal contribution

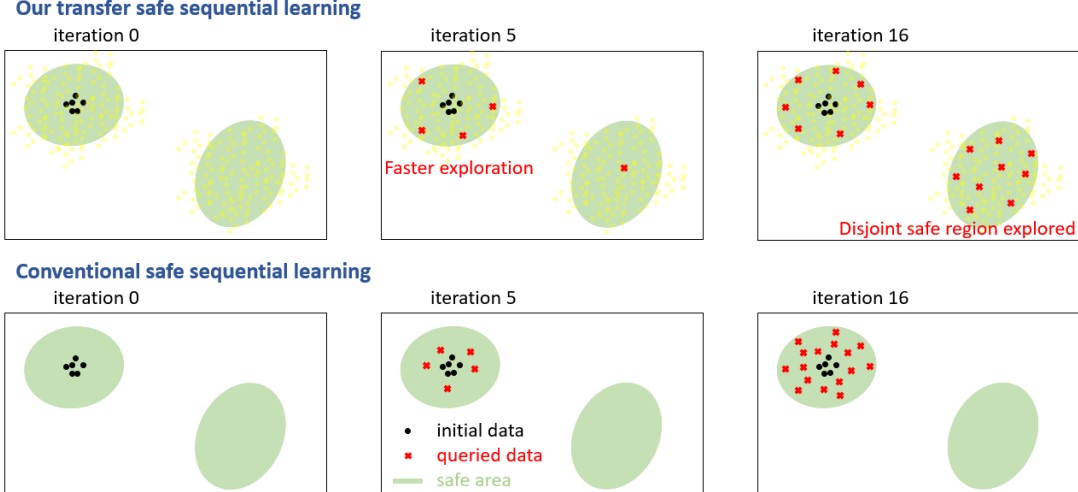

Figure 1: Illustration: Safe sequential learning with transfer (top) and conventional (bottom) learning. The light yellow data points represent source data. The main benefit of transfer learning is to accelerate exploration and identify larger and potentially disjoint safe regions by leveraging the source data.

on this information, receive observations from the oracle, and update the belief. This process is repeated until the learning goal is achieved, or until the acquisition budget is exhausted. These learning algorithms often utilize Gaussian processes (GPs, Rasmussen & Williams (2006)) as surrogate models for the acquisition computation (Krause et al., 2008; Brochu et al., 2010).

In many applications, such as spinal cord stimulation (Harkema et al., 2011) and robotic learning (Berkenkamp et al., 2016; Baumann et al., 2021), data acquisition can introduce safety risks due to unknown safety constraints in the input space. For instance, tuning a robot controller requires testing various controller parameters; however, certain parameter settings may lead to unsafe behaviors, such as a drone flying at high speed toward a human—an issue only observed after executing the controller (Berkenkamp et al., 2016). This scenario highlights the need for a safe learning approach that selects data points being safe and maximally informative within safety limits. One effective approach to safe learning is to model safety constraints using additional GPs (Sui et al., 2015; Schreiter et al., 2015; Zimmer et al., 2018; Sui et al., 2018; Turchetta et al., 2019; Berkenkamp et al., 2020; Sergeyev et al., 2020; Baumann et al., 2021; Li et al., 2022). These algorithms begin with a small set of safe observations, and define a safe set to restrict exploration to regions with high safety confidence. As learning progresses, this safe set expands, allowing the explorable area to grow over time. Safe learning approaches have also been explored in related fields, such as Markov Decision Processes (Turchetta et al., 2019) and reinforcement learning (García et al., 2015).

While safe learning methods have demonstrated significant impact, several challenges remain. First, the GP hyperparameters must be specified before exploration begins (Sui et al., 2015; Berkenkamp et al., 2016; 2020) or be fitted using an initially small dataset (Schreiter et al., 2015; Zimmer et al., 2018; Li et al., 2022). In addition, safe learning algorithms often suffer from local exploration: GP models are typically smooth, with uncertainty increasing beyond the boundaries of the reachable safe set. This results in slow convergence, and disconnected safe regions are often classified as unsafe and remain unexplored. We provide a detailed analysis and visual illustration of this issue in Section 4. In practice, local exploration complicates the deployment of safe learning algorithms, as domain experts must supply safe data from multiple distinct safe regions.

**Our contribution:** Safe learning generally begins with prior knowledge (Schreiter et al., 2015; Sui et al., 2015; Berkenkamp et al., 2020). We assume that correlated experiments have already been performed, and their results are readily available. This assumption enables transfer learning, offering two key benefits (see also Figure 1): (1) Exploration and expansion of safe regions are significantly accelerated, and (2)

disconnected safe regions can be explored allowing to discover larger safe regions. Both advantages are made possible by guidance from the source task. We empirically demonstrate both of the benefits and provide a theoretical analysis showing that conventional single-task approaches cannot identify unconnected safe regions. Real-world applications of this approach are ubiquitous, including simulation-to-reality transfer (Marco et al., 2017), serial production, and multi-fidelity modeling (Li et al., 2020).

Transfer learning can be implemented by jointly modeling the source and target tasks as multi-output GPs (Journel & Huijbregts, 1976; Álvarez et al., 2012). However, GPs are notorious for their cubic time complexity due to the inversion of Gram matrices (Section 3.1). Consequently, large volumes of source data significantly increase computational time, which is often a bottleneck in real-world experiments. To address this, we modularize the multi-output GPs, allowing source-related components to be precomputed and fixed, which reduces the computational complexity while retaining the benefits of transfer learning.

In summary, we 1) introduce the idea of safe transfer sequential learning, 2) derive that conventional single-task approaches cannot discover disjoint safe regions, 3) provide a modularized approach to multi-output GPs that alleviates the computational burden of incorporating the source data, and 4) demonstrate the empirical efficacy on safe AL problems.

**Related work:** Safe learning is considered in many applications including AL, BO, Markov Decision Processes (Turchetta et al., 2019) and reinforcement learning (García et al., 2015). In this paper, we focus on GP learning problems, as GPs are considered the gold-standard when it comes to calibrated uncertainties which is particularly important for safe learning under uncertainty. Previous works (Gelbart et al., 2014; Hernandez-Lobato et al., 2015; Hernández-Lobato et al., 2016) investigated constrained learning with GPs by incorporating constraints directly into the acquisition function (e.g., discounting the acquisition score by the probability of constraint violation). However, these approaches do not exclude unsafe data from the search pool, and generally address non-safety-critical applications. A safe set concept was introduced for safe BO (Sui et al., 2015) and safe AL (Schreiter et al., 2015), and later extended to BO with multiple safety constraints (Berkenkamp et al., 2020), to AL for time series modeling (Zimmer et al., 2018), and to AL for multi-output problems (Li et al., 2022). For safe BO, Sui et al. (2018) proposed a two-stage approach, separating safe set exploration and BO. However, all of these methods suffer from local exploration (Section 4). Some recent methods address disjoint safe regions. For example, Sergeyev et al. (2020) considered regions separated by small gaps where the constraint functions briefly fall below, but remain near, the safety threshold. Baumann et al. (2021) proposed a global safe BO method for dynamical systems, assuming that unsafe regions can be approached slowly enough such that an intervention mechanism exists to stop the system in time. Despite these advances, none of these approaches leverages safe transfer learning, which can allow for global exploration by utilizing prior knowledge from source tasks for a wide range of scenarios.

Transfer learning and multitask learning have gained increasing attention. In particular, multi-output GP methods have been developed for multitask BO (Swersky et al., 2013; Poloczek et al., 2017), sim-to-real transfer for BO (Marco et al., 2017), and multitask AL (Zhang et al., 2016). However, GPs face cubic time complexity with respect to the number of observations, a challenge that grows with multiple outputs. In Tighineanu et al. (2022), the authors assume a specific structure of the multi-output kernel, which allows to factorize the computation with an ensembling technique. This eases the computational burden for transfer sequential learning. In our paper, we propose a modularized safe transfer learning that avoids the cubic complexity.

**Paper structure:** The remaining of this paper is structured as follows. We provide the setup and problem statement in Section 2, background on GPs and safe AL in Section 3. Section 4 discusses theoretical perspective of safe learning and demonstrate that safe learning approaches based on standard GPs suffer from local exploration. Section 5 elaborates our safe transfer learning approach and our modular computation scheme. Section 6 is the experimental study. Finally, we conclude our paper in Section 7.

Table 1: Key Notation

| Symbols | Meaning |
|---|---|
| $N_{\text{init}}$ | number of initial target data points |
| $N_{\text{query}}$ | number of target data points added by AL |
| $N = N_{\text{init}}, \dots, N_{\text{init}} + N_{\text{query}}$ | number of total target points |
| $N_{\text{source}}$ | number of source data points |
| $\mathcal{D}_N = \{\boldsymbol{x}_{1:N}, y_{1:N}, \boldsymbol{z}_{1:N}\}$ | dataset of the target task |
| $\mathcal{D}_{N_{\text{source}}}^{\text{source}} = \{\boldsymbol{x}_{s,1:N_{\text{source}}}, y_{s,1:N_{\text{source}}}, \boldsymbol{z}_{s,1:N_{\text{source}}}\}$ | dataset of the source task |
| $\boldsymbol{z} = (z^1, \dots, z^J)$ | safety variables of the target task |
| $\boldsymbol{z}_s = (z_s^1, \dots, z_s^J)$ | safety variables of the source task |
| $y = f(\boldsymbol{x}) + \epsilon_f$ | model of the target observation $y$ |
| $z^j = q^j(\boldsymbol{x}) + \epsilon_{q^j}$ | model of the target safety observation $z^j$ |
| $y_s = f_s(\boldsymbol{x}) + \epsilon_{f_s}$ | model of the source observation $y_s$ |
| $z_s^j = q_s^j(\boldsymbol{x}) + \epsilon_{q_s^j}$ | model of the source safety observation $z_s^j$ |
| $f \sim \mathcal{GP}\left(\boldsymbol{0}, k_f\right)$ | single-output GP prior over target main function $f$ |
| $q^j \sim \mathcal{GP}\left(\boldsymbol{0}, k_{q^j}\right)$ | single-output GP prior over target safety function $q^j$ |
| $\boldsymbol{f} \sim \mathcal{GP}\left(\boldsymbol{0}, k_{\boldsymbol{f}}\right)$ | multi-output GP prior over main functions $f_s$ and $f$ |
| $\boldsymbol{q}^j \sim \mathcal{GP}\left(\boldsymbol{0}, k_{\boldsymbol{q}^j}\right)$ | multi-output GP prior over safety functions $q_s^j$ and $q^j$ |

## 2 Safe Transfer Active Learning Setup

Transfer Learning aims to transfer knowledge from previous, *source*, systems to a new, *target*, system. Usually, there exist a lot of data from one or more source systems and only few or no data from the target system. Safe Transfer Active Learning will exploit the knowledge from the source systems' data and allows for safe and active data collection on the target system. Throughout this paper, we inspect regression problems.

**Target and Safety – Notation:** Each $D$-dimensional input $\boldsymbol{x} \in \mathcal{X} \subseteq \mathbb{R}^D$ has a corresponding regression output $y \in \mathbb{R}$ and safety values jointly expressed as $\boldsymbol{z} = (z^1, \dots, z^J) \in \mathbb{R}^J$, $J$ is the number of safety variables. There are $J$ thresholds $T_j \in \mathbb{R}, j = 1, \dots, J$, and an input $\boldsymbol{x}$ is safe if the corresponding safety values $z^j \geq T_j$ for all $j = 1, \dots, J$. It is assumed that the underlying functions of $y, z^1, \dots, z^J$ are all unknown.

**Source and Safety – Notation:** Similarly, there exist output and safety values of one or more source tasks, again from unknown underlying functions. The source output value is denoted by $y_s \in \mathbb{R}$ and source safety values by $\boldsymbol{z}_s = (z_s^1, \dots, z_s^J) \in \mathbb{R}^J$, $s$ is the index of source task(s). The source tasks are defined on the same domain $\mathcal{X}$. The source and target tasks may have different numbers of constraint variables, but we can add trivial constraints (e.g. $1 \geq -\infty$) to any of the tasks in order to have the same number $J$. Furthermore, the source data may or may not be measured with the same safety constraints as the target task. For example, in a simulation-to-reality transfer (Marco et al., 2017), the source dataset can be obtained unconstrained.

**Datasets – Notation:** A dataset over the target task is denoted by $\mathcal{D}_N = \{\boldsymbol{x}_{1:N}, y_{1:N}, \boldsymbol{z}_{1:N}\}$, $\boldsymbol{x}_{1:N} = \{\boldsymbol{x}_1, \dots, \boldsymbol{x}_N\} \subseteq \mathcal{X}$, $y_{1:N} = \{y_1, \dots, y_N\} \subseteq \mathbb{R}$, safety observations $\boldsymbol{z}_{1:N} := \{\boldsymbol{z}_n = (z_n^1, \dots, z_n^J)\}_{n=1}^N \subseteq \mathbb{R}^J$, and $N$ is the number of observed data. In this paper, $N$ is not fixed, as we may actively add new labeled data. We denote the source data by $\mathcal{D}_{N_{\text{source}}}^{\text{source}} = \{\boldsymbol{x}_{s,1:N_{\text{source}}}, y_{s,1:N_{\text{source}}}, \boldsymbol{z}_{s,1:N_{\text{source}}}\} \subseteq \mathcal{X} \times \mathbb{R} \times \mathbb{R}^J$, $s$ is the index of source task and $N_{\text{source}}$ is the number of all source data points. In our main paper, we consider only one source task for simplicity, while Appendix D.2 provides formulation on more source tasks. Please also see Table 1 for a summary of our notation.

**Safe Active Learning Procedure:** The goal of safe AL is to collect data actively and safely on the target system, such that the final dataset helps to model the regression output $y$ on the safe region of input space $\mathcal{X}$, i.e. subset of $\mathcal{X}$ corresponding to $z^1 \geq T_1, \dots, z^J \geq T_J$.

Concretely speaking, we are given a small amount of data on the target task, i.e. $\mathcal{D}_N$ where the initial size $N = N_{\text{init}}$ is small. The initial data are typically given by a domain expert and are safe, i.e. for $\mathcal{D}_{N_{\text{init}}} = \{\boldsymbol{x}_{1:N_{\text{init}}}, y_{1:N_{\text{init}}}, \boldsymbol{z}_{1:N_{\text{init}}}\}, \forall n = 1, ..., N_{\text{init}}, \boldsymbol{z}_n = (z_n^1, ..., z_n^J)$ satisfy the safety constraints $z_n^1 \geq T_1, ..., z_n^J \geq T_J$.

At each $N$, one seeks the next point $\boldsymbol{x}_* \in \mathcal{X}_{\text{pool}} \subseteq \mathcal{X}$ to be evaluated. $\mathcal{X}_{\text{pool}} \subseteq \mathcal{X}$ is the search pool which can be the entire space $\mathcal{X}$ or a predefined subspace of $\mathcal{X}$, depending on the applications. The evaluation is budget consuming and safety critical, and it will return a noisy $y_*$ and noisy safety values $\boldsymbol{z}_*$. Ideally, we need to make sure that $\boldsymbol{z}_* = (z_*^1, ..., z_*^J)$ respect the safety constraints $z_*^j \geq T_j$ for all $j = 1, ..., J$ and that $y_*$ is informative for the modeling of target $y$. As the safety outputs are unknown when an $\boldsymbol{x}_*$ is selected, guaranteeing safety is challenging. Safe learning methods resort to allowing queries that are safe only with high probability (Sui et al., 2015; 2018; Zimmer et al., 2018; Li et al., 2022).

Afterward, the labeled point is added to $\mathcal{D}_N$ (observed dataset becomes $\mathcal{D}_{N+1}$), and we proceed to the next iterations. $N$ is initially $N_{\text{init}}$ and grows to $N_{\text{init}} + N_{\text{query}}$. $N_{\text{query}}$ is the number of the learning iterations, i.e. the number of data points actively added.

**Safe Transfer Active Learning Aim:** In particular, this paper aims to build a new safe transfer AL, a safe AL algorithm with multi-output GPs, so that we leverage the information of the source data $\mathcal{D}_{N_{\text{source}}}^{\text{source}}$ to explore a larger safe area. Our algorithm aims to

- (i) collect as few (small $N_{\text{query}}$) data as possible for building an accurate regression model of $y$ (in the safe part of the input domain $\mathcal{X}$),

- (ii) collect the data in a safe way and hereby explore the safe region including its boundaries,

- (iii) in particular explore larger safe areas than benchmarks in a faster way.

## 3 Background: Gaussian Processes of Single-Task, Safe Active Learning

In this section, we introduce Gaussian Processes (GPs) and state-of-the-art safe Active Learning (safe AL). GPs are the workhorse of safe AL in which they are routinely applied to select safe and informative data points (Schreiter et al., 2015; Zimmer et al., 2018; Li et al., 2022).

### 3.1 Gaussian Processes (GPs)

Suppose we aim to model the output $y$ and the safety observations $z^1, ..., z^J$ with GPs. Here, we introduce the modeling scheme and the underlying assumptions. The first assumption is that the data represent functional values blurred with i.i.d. Gaussian noises.

**Assumption 3.1** (Data: target task)**.** Assume $y = f(\boldsymbol{x}) + \epsilon_f$, where $\epsilon_f \sim \mathcal{N}\left(0, \sigma_f^2\right)$, for our target observations. We further assume that $z^j = q^j(\boldsymbol{x}) + \epsilon_{q^j}$, where $\epsilon_{q^j} \sim \mathcal{N}\left(0, \sigma_{q^j}^2\right)$, and $j = 1, \ldots, J$ indexes the safety constraints. All of the noise variances $\{\sigma_f^2, \sigma_{q^1}^2, \ldots, \sigma_{q^J}^2\}$ are positive.

We then place a GP assumption on each of the underlying functions $f, q^1, ..., q^J$. A GP is a stochastic process defined by a mean and a kernel function (Rasmussen & Williams, 2006; Kanagawa et al., 2018; Schoelkopf & Smola, 2002). In this work, we set the mean to zero — a common practice, as normalized data typically justifies this assumption. The kernel function, $\mathcal{X} \times \mathcal{X} \to \mathbb{R}$, specifies the covariance of function values at different input points. Without prior knowledge of the data, we make the standard assumption that the governing kernels are stationary. The GP assumption is then formulated as the following.

**Assumption 3.2** (Model: single-task)**.** For each function, $g \in \{f, q^1, ..., q^J\}$, we assume that $g \sim \mathcal{GP}(0, k_g)$ with a stationary kernel with bounded variance, $k_g(\boldsymbol{x}, \boldsymbol{x}') := k_g(\boldsymbol{x} - \boldsymbol{x}') \leq 1$.

Bounding the kernels by one provides advantages in theoretical analysis (Srinivas et al., 2012) and is not restrictive since the data is usually normalized to unit variance.

Assumption 3.1 and Assumption 3.2 provide the predictive distribution of the functions $f, q^1, ..., q^J$. We write down the distribution for the function $f$ at a test point $\boldsymbol{x}_*$:

$$p\left(f(\boldsymbol{x}_*)|\boldsymbol{x}_{1:N}, y_{1:N}\right) = \mathcal{N}\left(\mu_{f,N}(\boldsymbol{x}_*), \sigma_{f,N}^2(\boldsymbol{x}_*)\right), \tag{1}$$

where

$$
\begin{aligned}
\mu_{f,N}(\boldsymbol{x}_*) &= k_f(\boldsymbol{x}_{1:N}, \boldsymbol{x}_*)^T \left(\boldsymbol{K}_f + \sigma_f^2 I\right)^{-1} y_{1:N}, \\
\sigma_{f,N}^2(\boldsymbol{x}_*) &= k_f(\boldsymbol{x}_*, \boldsymbol{x}_*) - k_f(\boldsymbol{x}_{1:N}, \boldsymbol{x}_*)^T \left(\boldsymbol{K}_f + \sigma_f^2 I\right)^{-1} k_f(\boldsymbol{x}_{1:N}, \boldsymbol{x}_*).
\end{aligned}
\tag{2}
$$

We use the notation $k_f(\boldsymbol{x}_{1:N}, \boldsymbol{x}_*) = (k_f(\boldsymbol{x}_1, \boldsymbol{x}_*), ..., k_f(\boldsymbol{x}_N, \boldsymbol{x}_*)) \in \mathbb{R}^{N \times 1}$ to denote the kernel vector between the training points $\boldsymbol{x}_{1:N}$ and the test point $\boldsymbol{x}_*$. The kernel matrix $\boldsymbol{K}_f \in \mathbb{R}^{N \times N}$ contains the covariances between the training points $\boldsymbol{x}_{1:N}$ with $[\boldsymbol{K}_f]_{m,n} = k_f(\boldsymbol{x}_m, \boldsymbol{x}_n), m, n = 1, ..., N$.

Typically, $k_f$ is parameterized and can be jointly fitted with the noise variance $\sigma_f^2$. Common fitting techniques involve computing the marginal likelihood, $\mathcal{N}\left(y_{1:N}|\boldsymbol{0}, \boldsymbol{K}_f + \sigma_f^2 I\right)$, where the the runtime complexity is $\mathcal{O}\left(N^3\right)$, dominated by the inversion of the Gram matrix $\left(\boldsymbol{K}_f + \sigma_f^2 I\right)^{-1}$.

The predictive distributions of the safety functions $q^1, ..., q^J$ can be obtained by replacing $f$ with $q^1, ..., q^J$ and the outputs $y_{1:N}$ with $z_{1:N}^j, j = 1, ..., J$ in Equations (1) and (2). Similarly, the log-likelihood can be maximized for each $q^j$ by jointly learning $k_{q^j}$ and $\sigma_{q^j}^2$ in the same manner, $j = 1, \ldots, J$.

**Remark 3.3.** In our paper, all safety measurements $z^1, ..., z^J$ are modeled independently. If the variables are not independent, our analysis and arguments still apply, as the dependent constraints can be grouped, and the problem reduces back to the independent case.

## 3.2 Safe Active Learning (Safe AL)

In this section, we introduce state-of-the-art safe AL. The state-of-the-art safe AL cannot exploit knowledge in form of source data. Although source data is not considered in this section, we will still write *target task* to make the distinction between the two tasks clear.

Safe AL (Schreiter et al., 2015; Zimmer et al., 2018; Li et al., 2022) aims to select data actively and safely to learn about a target task. At a given number of available target data $N$, the goal is to select $\boldsymbol{x}_* \in \mathcal{X}_{\text{pool}} \subseteq \mathcal{X}$, that gives us a safe $\boldsymbol{z}_* = (z_*^1, ..., z_*^J)$ and informative $y_*$ (informative in the context of modeling).

The key of safe AL is the safety and data selection criteria. Commonly, these criteria employ GPs due to the well calibrated predictive uncertainty (Schreiter et al., 2015; Zimmer et al., 2018; Li et al., 2022). It is worth highlighting a closely related field, safe Bayesian optimization (BO) (Sui et al., 2015; 2018; Berkenkamp et al., 2020), which follows a similar procedure except that the goal is to search for an optimum $y_*$ subject to safety constraints $z_*^1 \geq T_1, ..., z_*^J \geq T_J$.

**Safety Condition:** The safety variables are modeled by GP functions $q^1, ..., q^J$ (Section 3.1), and the core is to compare the safety confidence bounds (Equation (2)) with the thresholds $T_1, ..., T_J \in \mathbb{R}$. At each

---

**Algorithm 1** Safe AL

---

**Require:** Assumption 3.1, Assumption 3.2, $\mathcal{D}_{N_{\text{init}}}, \mathcal{X}_{\text{pool}}, \beta$ or $\alpha$, $N_{\text{query}}$, thresholds $T_1, ..., T_J$
1: **for** $N = N_{\text{init}}, ..., N_{\text{init}} + N_{\text{query}} - 1$ **do**
2:      Fit GPs $f, q_1, ..., q_J$ with $\mathcal{D}_N$
3:      $\mathcal{S}_N \leftarrow \cap_{j=1}^J \{\boldsymbol{x} \in \mathcal{X}_{\text{pool}} | \mu_{q^j,N}(\boldsymbol{x}) - \beta^{1/2}\sigma_{q^j,N}(\boldsymbol{x}) \geq T_j\}$ (Equations (2) and (3))
4:      $\boldsymbol{x}_* \leftarrow \text{argmax}_{\boldsymbol{x} \in \mathcal{S}_N} a(\boldsymbol{x}), \ a(\boldsymbol{x}) = \sum_{g \in \{f, q^1, ..., q^J\}} H_g[\boldsymbol{x}|\mathcal{D}_N]$
5:      Query $\boldsymbol{x}_*$ to get evaluations $y_*$ and $\boldsymbol{z}_*$
6:      $\mathcal{D}_{N+1} \leftarrow \mathcal{D}_N \cup \{\boldsymbol{x}_*, y_*, \boldsymbol{z}_*\}, \mathcal{X}_{\text{pool}} \leftarrow \mathcal{X}_{\text{pool}} \setminus \{\boldsymbol{x}_*\}$
7: **end for**
8: **Return** $\mathcal{D}_{N_{\text{init}}+N_{\text{query}}}$, trained models $f, q_1, ..., q_J$

---

iteration $N$, we can compute the safety probability $p\left(q^j(\boldsymbol{x})|\boldsymbol{x}_{1:N}, z^j_{1:N}\right) = \mathcal{N}\left(\mu_{q^j,N}(\boldsymbol{x}), \sigma^2_{q^j,N}(\boldsymbol{x})\right)$, for each safety constraint $j = 1, ..., J$. Sui et al. (2015) defines the safe set $\mathcal{S}_N \subseteq \mathcal{X}_{\text{pool}}$ as

$$\mathcal{S}_N = \cap_{j=1}^J \{\boldsymbol{x} \in \mathcal{X}_{\text{pool}} | \mu_{q^j,N}(\boldsymbol{x}) - \beta^{1/2}\sigma_{q^j,N}(\boldsymbol{x}) \geq T_j\}, \tag{3}$$

where $\beta \in \mathbb{R}^+$ is a parameter for probabilistic tolerance control. This definition is equivalent to $\forall \boldsymbol{x} \in \mathcal{S}_N, p\left(q^1(\boldsymbol{x}) \geq T_1, ..., q^J(\boldsymbol{x}) \geq T_J\right) \geq (1-\alpha)^J$ when $\alpha = 1 - \Phi(\beta^{1/2})$ ($\alpha \in [0,1]$) where $\Phi$ is the standard Gaussian cumulative distribution function (CDF) (Schreiter et al., 2015; Zimmer et al., 2018; Li et al., 2022). Note that $\alpha > 0.5$ corresponds to $\beta^{1/2} < 0$ (assume this exists) which is usually not considered because safe learning aims for high safety confidence while $\alpha > 0.5$ indicates a safety confidence of at most 50% per constraint - so at most a random guess.

**Information Criterion:** Safe AL queries a new point by mapping safe candidate inputs to acquisition scores:

$$\boldsymbol{x}_* = \text{argmax}_{\boldsymbol{x} \in \mathcal{S}_N} a\left(\boldsymbol{x}\right), \tag{4}$$

where $a(\cdot)$ is an acquisition function. Notice here that $a(\cdot)$ and $\mathcal{S}_N$ both depend on the observed dataset $\mathcal{D}_N$. In AL problems, a prominent acquisition function is the predictive entropy (Schreiter et al., 2015; Zimmer et al., 2018; Li et al., 2022): $a(\boldsymbol{x}) = H_f\left[\boldsymbol{x}|\mathcal{D}_N\right] = \frac{1}{2}\log\left(2\pi e\sigma^2_{f,N}(\boldsymbol{x})\right)$, where $\sigma^2_{f,N}$ is defined in Equation (2). To accelerate the exploration of the safety functions, Berkenkamp et al. (2020) incorporate the information of the safety functions by using $a(\boldsymbol{x}) = \max_{g \in \{f, q^1, ..., q^J\}}\{H_g\left[\boldsymbol{x}|\mathcal{D}_N\right]\}$. Our acquisition function is built upon this and is written in Algorithm 1.

Please note again the close connection to BO: it is possible to exchange the acquisition function by the SafeOpt criteria if one wants to address safe BO problems (Sui et al., 2015; Berkenkamp et al., 2020; Rothfuss et al., 2022)).

**Constrained Acquisition Optimization:** Solving Equation (4) is challenging. In the literature (Schreiter et al., 2015; Li et al., 2022; Sui et al., 2015; Berkenkamp et al., 2020), this is solved on a discrete pool with finite elements, i.e. $N_{\text{pool}} := |\mathcal{X}_{\text{pool}}| < \infty$. One would compute the GP posteriors on the entire pool $\mathcal{X}_{\text{pool}}$ to determine the safe set, then optimize the acquisition scores over the safe set. In our paper, we inherit this finite discrete pool setting.

**Time Complexity:** The complexity comprises $\mathcal{O}\left(N^3\right)$ for GP training and $\mathcal{O}\left(N_{\text{pool}}N^2\right)$ for GP inference, assuming the Gram matrices, $\left(\boldsymbol{K}_g + \sigma^2_g I\right)^{-1}, \forall g \in \{f, q^1, ..., q^J\}$, are already computed during the training (Equation (2)). Importantly, GP inference is only performed once per query, whereas GP training (or more specifically, its most computationally expensive step, the matrix inversion) is repeated multiple times during parameter learning. As a consequence, the size of the discretized pool $N_{\text{pool}}$ can be much larger than the training dataset $N$, e.g. $\mathcal{X}_{\text{pool}}$ can include up to tens or even hundreds of thousands of samples.

The whole learning process is summarized in Algorithm 1. In the next section, we provide theoretical analysis to demonstrate the presence of local exploration in safe learning approaches.

# 4 Safe Learning Solely on Target Task: Local Exploration

Before we introduce our safe transfer AL approach in the next Section 5, we analyze a shortcoming of the standard safe AL (Algorithm 1). We quantify the upper bound for an explorable safe region, and prove that safe AL is limited to local exploration within the given bound. Note, that the analysis will not involve the acquisition function, and therefore the result applies not only to safe AL but also to GP based safe BO settings with safe set as in Equation (3).

Given observations $\mathcal{D}_N$, we would like to know, until how far into the input space the safety confidence is sufficiently high.

**Correlation weakened with increasing distance:** The conventional safe AL (Algorithm 1) builds models based on a standard GP assumption (Assumption 3.1, Assumption 3.2), and then the explorable region is obtained by quantifying the safety confidence, conditioned on observed data $\mathcal{D}_N$ (Equation (3)). The safety confidence is calculated from the GP predictive distributions (Equation (2)), and it thus depends on the kernel to correlate input points of various locations. Commonly used stationary kernels measure the distance between a pair of points, while the actual output values do not matter (for two points $\boldsymbol{x}, \boldsymbol{x}' \in \mathcal{X}$, $\|\boldsymbol{x} - \boldsymbol{x}'\|$ determines the covariance). These kernels have the property that closer points have higher kernel values, indicating stronger correlation, while distant points result in small kernel values. We first formulate this property as the following.

**Definition 4.1.** We call a kernel $k$ a kernel with *correlation weakened by distance* if $k : \mathcal{X} \times \mathcal{X} \to \mathbb{R}$ fulfills the following property: $\forall \delta > 0, \exists r > 0$ s.t. $\|\boldsymbol{x} - \boldsymbol{x}'\| \geq r \Rightarrow k(\boldsymbol{x}, \boldsymbol{x}') \leq \delta$ under $L2$ norm.

This definition will later help us quantify the upper bound of an explorable region. We provide expressions of popular stationary kernels (RBF kernel and Matérn kernels), as well as their relations between input distance $r$ and covariance $\delta$ in the Appendix B.

**Low correlation leading to small safety probability:** With Definition 4.1, we can now derive a theorem measuring the explorable region. The main idea is: when an unlabeled point $\boldsymbol{x}_*$ is far away from the observed inputs, the value of $\delta$ can become very small (i.e. small covariance measured by the kernel). Thus, for each $j = 1, ..., J$, the model weakly correlates $q^j(\boldsymbol{x}_*)$ to the observations. As a result, the predictive mean is close to zero (GP prior) and the predictive uncertainty is large, both of which imply that the method has small safety confidence,[1] at least if $\forall j = 1, ..., J, q^j \geq T_j$ is not a trivial condition, e.g. a trivial condition would be if a function $q^j$ has values majorly distributed in $[-1, 1]$ but thresholded at $T_j = -2$.

**Theorem 4.2** (Local exploration of single-output GPs). *We are given $\boldsymbol{x}_{1:N} \subseteq \mathcal{X}$. For any safety constraint indexed by $j = 1, ..., J$, let $z_{1:N}^j := (z_1^j, ..., z_N^j)$ be the observed noisy safety values and let $\|(z_1^j, ..., z_N^j)\| \leq \sqrt{N}$. The safety value $z^j = q^j(\boldsymbol{x}) + \epsilon_{q^j}$ satisfies the GP assumptions (Assumption 3.1, Assumption 3.2): $q^j \sim \mathcal{GP}(0, k_{q^j}), k_{q^j}(\cdot, \cdot) \leq 1, \epsilon_{q^j} \sim \mathcal{N}\left(0, \sigma_{q^j}^2\right)$. The kernel $k_{q^j}$ is a kernel with correlation weakened by distance (Definition 4.1). Denote $k_{scale}^j := max\ k_{q^j}(\cdot, \cdot)$. Then $\forall \delta \in (0, \sqrt{k_{scale}^j} \sigma_{q^j}/\sqrt{N}), \exists r > 0$ s.t. $\forall \boldsymbol{x}_* \in \mathcal{X}$ that fulfill $min_{\boldsymbol{x}_i \in \boldsymbol{x}_{1:N}} \|\boldsymbol{x}_* - \boldsymbol{x}_i\| \geq r$, the probability thresholded on a constant $T_j$ is bounded by $p\left((q^j(\boldsymbol{x}_*) \geq T_j)|\boldsymbol{x}_{1:N}, z_{1:N}^j\right) \leq \Phi\left(\frac{N\delta/\sigma_{q^j}^2 - T_j}{\sqrt{k_{scale}^j - (\sqrt{N}\delta/\sigma_{q^j})^2}}\right)$. $\Phi$ is the CDF of standard Gaussian.*

To prove this theorem, we need Equation (2) to compute the safety likelihood, and then we use the eigenvalues of the kernel Gram matrix together with Definition 4.1 to derive the final bound (see Appendix C for the detailed proof).

Our theorem provides the maximum safety probability of a point as a function of its distance to the observed data in $\mathcal{X}$. The safe set tolerance parameter $\beta$ or $\alpha$ (Equation (3)) can be used to compute the covariance bound $\delta$. For example, when $J = 1$ and $\beta^{1/2} = 2$, which means $p(q(\boldsymbol{x}) \geq T) \geq \Phi(2)$ is safe (Equation (3)), we choose a $\delta$ such that $\Phi\left(\frac{N\delta/\sigma_q^2 - T}{\sqrt{k_{scale} - (\sqrt{N}\delta/\sigma_q)^2}}\right) \leq \Phi(2)$ ($j$ omitted when $J = 1$). Such a $\delta$ exists in all situation of our interest, as we will soon discuss. Given a $\delta$, we can then determine a corresponding radius $r$ (see e.g. Appendix B). Interpreting $r$ as the radius around the observed data, the safety confidence outside this region always remains low. Since safety confidence decides the explorable regions (Equations (3) and (4)), this theorem measures an upper bound of explorable safe area. The upper bound is given for one safety constraint, and we can see from Equation (3) that the final bound of safety confidence is the product of the $\Phi$ term over different $j$. In other words, the more safety constraints, the smaller the explorable regions may be, which is intuitive.

---

[1] A small safety confidence indicates $\exists j = 1, ..., J, p\left((q^j(\boldsymbol{x}_*) \geq T_j)|\boldsymbol{x}_{1:N}, z_{1:N}^j\right)$ is not high enough.

Notice that $\|(z_1^j, ..., z_N^j)\| \leq \sqrt{N}$ $(\forall j = 1, ..., J)$ is not very restrictive because an unit-variance dataset has $\|(z_1^j, ..., z_N^j)\| = \sqrt{N}$. Note further that $\delta \in \left(0, \sqrt{k_{scale}^j}\sigma_{q^j}/\sqrt{N}\right)$ implies $k_{scale}^j > (\sqrt{N}\delta/\sigma_{q^j})^2$, which means the bound $\Phi\left(\frac{N\delta/\sigma_{q^j}^2 - T_j}{\sqrt{k_{scale}^j - (\sqrt{N}\delta/\sigma_{q^j})^2}}\right)$ is always valid.

This theorem indicates that a standard GP with commonly used kernels explores only neighboring regions of the initial $\boldsymbol{x}_{1:N}$. The theorem is independent of the acquisition functions, and thus the local exploration problems present in all safe learning methods based on standard GPs.

**Existence of $\delta$ for common safe learning situations:** We would like to illustrate an example of using our theorem to compute an explorable bound. Before that, we will make a statement relating the safety level $\beta$ to the quantity $\delta$ used in Theorem 4.2. This shows that a $\delta$ and, therefore, local exploration is present in all but some (at least) uncommon scenarios, which are in fact out of interest for the sake of safe exploration.

**Corollary 4.3** (Existence of $\delta$). *We are given the assumptions in Theorem 4.2. For each $j = 1, ..., J$, if either (1) $T_j \geq 0, \beta^{1/2} > 0$ or (2) $T_j < 0, \beta^{1/2} > \frac{|T_j|}{\sqrt{k_{scale}^j}}$, then $\exists \delta \in \left(0, \sqrt{k_{scale}^j}\sigma_{q^j}/\sqrt{N}\right)$ s.t.*

$$\Phi\left(\frac{N\delta/\sigma_{q^j}^2 - T_j}{\sqrt{k_{scale}^j - (\sqrt{N}\delta/\sigma_{q^j})^2}}\right) \leq \Phi(\beta^{1/2}).$$

This corollary can be proved by inspecting the boundary of each constant (detailed in Appendix C).

The key insight is that, a $\delta$ in Theorem 4.2, which bounds the safety likelihood, always exists for common selection of safety level $\beta^{1/2}$. There are two scenarios considered in Corollary 4.3. The first scenario, $T_j \geq 0, \beta^{1/2} > 0$ is common because $\beta^{1/2} > 0$ is always desired for safe exploration and stricter safety thresholds $T_1 \geq 0, ..., T_J \geq 0$ may also occur. In the second scenario, the thresholds are softer, i.e. one or more of the thresholds $T_1, ..., T_J$ are smaller than zero. It turns out that $\beta^{1/2} > |T_j|/\sqrt{k_{scale}^j}, j = 1, ..., J$ is desired as well for safe learning. Consider $|T_j|/\sqrt{k_{scale}^j} \geq \beta^{\frac{1}{2}} > 0$ for a $j \in \{1, ..., J\}$, which is the scenario not fulfilling the condition. We focus on normalized variable $z_j$, where the underlying function is modeled by a GP $q^j \sim \mathcal{GP}(0, k_{q^j})$. When this model extrapolates in regions where data are absent, the inference is highly based on the prior $q^j(\boldsymbol{x}) \sim \mathcal{N}(0, k_{scale}^j)$. The safe set considers $p(q^j(\boldsymbol{x}) \geq T_j) \geq \Phi(\beta^{1/2})$ as a safety condition on the $j$-th constraint, but the prior indicates $\Phi(-T_j/\sqrt{k_{scale}^j}) \geq \Phi(\beta^{1/2})$ which becomes a trivial condition when $|T_j|/\sqrt{k_{scale}^j} \geq \beta^{\frac{1}{2}} > 0$. In other words, any input $\boldsymbol{x}$ has a safe prior unless the data disagree. This is a scenario that is not of interest for safe learning.

Therefore, for all common selection of safety level $\beta^{1/2}$, Corollary 4.3 implies that we can find a $\delta$ and apply Theorem 4.2 to quantify the upper bound of explorable safe set, which shows the presence of local exploration.

**Illustrating the theoretical result:** In the following, we plug exact numbers into Theorem 4.2 for an illustration.

**Example 4.4.** We consider a one-dimensional toy dataset visualized in Figure 2. Assume $N = 10$, $\sigma_q^2 = 0.01$ and constraint $T = 0$. We omit safety constraint index $j$, since $J = 1$ in this case. In this example, the generated data have $\|z_{1:N}\| \leq \sqrt{10}$. $\sigma_q/\sqrt{N}$ is roughly 0.0316. We train an unit-variance ($k_{scale} = max \ k_q(\cdot, \cdot) = 1$, theorem requires $\delta < 0.0316$) Matérn-5/2 kernel on this example, resulting in a learned lengthscale of around 0.1256. The Matérn-5/2 kernel is a kernel with correlation weakened by distance (Definition 4.1). In particular, $r = 4.485 * 0.1256 = 0.563316 \Rightarrow \delta \leq 0.002$ (Appendix B), noticing that $\delta = 0.002 \Rightarrow \Phi\left(\frac{N\delta/\sigma_q^2 - T}{\sqrt{1 - (\sqrt{N}\delta/\sigma_q)^2}}\right) \approx \Phi(2)$. Consider $\beta^{1/2} = 2$, then it is safe only when the safety likelihood is above $\Phi(2)$. We can thus know from Theorem 4.2 that safe regions that are 0.563316 further from the observed ones are always identified as unsafe and is not explorable. In Figure 2, the two safe regions are more than 0.7 distant from each other, indicating that the right safe region is never explored by conventional safe learning methods.

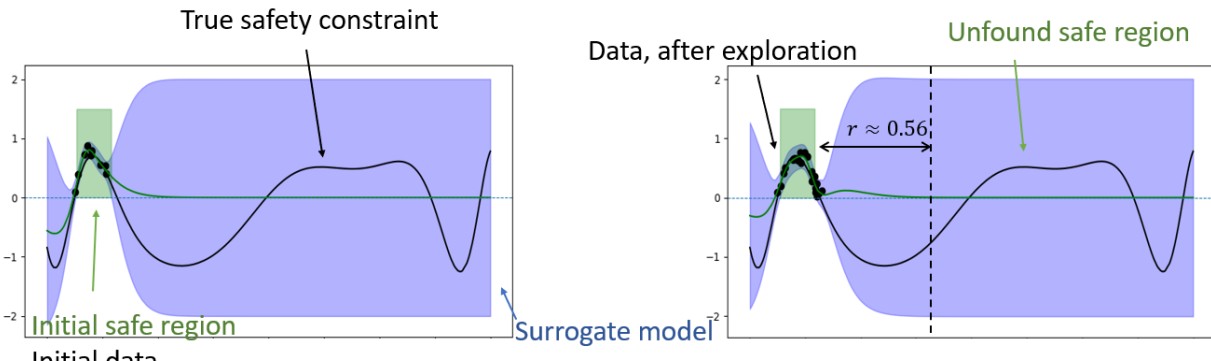

Figure 2: A safety function (shown in black) with two safe regions above threshold zero. Left graphics: Based on the initial data within one of the safe regions, a GP surrogate is trained. The blue line represents the mean prediction, while the blue shaded area indicates the uncertainty (e.g., confidence interval) around the mean. The green area indicates the learned safe area. Right graphics: After exploration, more points are sampled within the first safe region. However, the gap to the second safe region exceeds $r$, preventing the discovery of the second region, rendering the learned safe area almost unchanged. The true safety function used here is $q(x) = \sin\left(10x^3 - 5x - 10\right) + \frac{1}{3}x^2 - \frac{1}{2}$. The observations are with noise drawn from $\mathcal{N}(0, 0.1^2)$.

**GP might even explore less in practice:** Our probability bound $\Phi\left(\frac{N\delta/\sigma_{q^j}^2 - T_j}{\sqrt{k_{scale}^j - (\sqrt{N}\delta/\sigma_{q^j})^2}}\right)$ (for each $j = 1, ..., J$) is the worst case obtained with very mild assumptions. Empirically, the explorable regions found by GP models are smaller (see Figure 2).

**Transfer learning may overcome the local exploration:** We extended the Example 4.4 to compare the standard GP model against a transfer task GP which will be introduced in the next Section 5. In Figure 3, a linear model of corregionalization is trained (a kind of multitask GP, Álvarez et al. (2012)). On the right region, which is beyond the explorable bound of a standard single-task GP, the transfer task GP incorporates the source data allowing to build high safety confidence. As a result, the right region can be included into the explorable safe set (detailed in the following Section 5). We will also confirm in our experiments in Section 6 that guidance from source data enables our new safe transfer AL framework to explore beyond the immediate neighborhood of the target points $\boldsymbol{x}_{1:N}$.

To summarize, we see in this section that the safe set of standard GPs (Equation (3)) is limited to a local region. In the next section, we transfer knowledge from the source data to expand the exploration beyond the seed dataset of the target task.

## 5 Safe Transfer Active Learning and Source Pre-Computation

In this section, we formulate our safe transfer AL method. We start from introducing transfer task GPs, then we leverage the source data and the transfer task GPs to adapt Algorithm 1. We state the resulting new constrained optimization problem for safe transfer AL. We then explain the complexity and consider a modular computation to facilitate the algorithm. We conclude the section by describing our kernel choices for the experiments.

### 5.1 Background: Transfer Task GPs

In the presence of a source task, one can model the source task and the target task jointly with multi-output GPs (Journel & Huijbregts, 1976; Álvarez et al., 2012; Tighineanu et al., 2022). The key idea is to augment the input with a task index variable, allowing the model to distinguish between tasks while sharing information across them. Leveraging a source task in this way improves data efficiency on the target system, as relevant

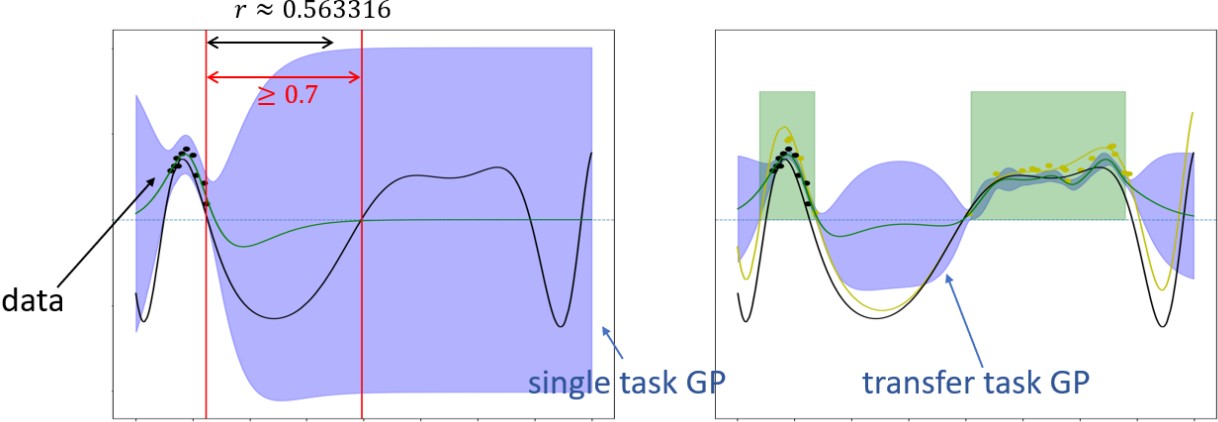

Figure 3: The same safety constraint as in Figure 2 with two safe regions. Left: the single-task GP cannot reach the right safe region as the distance is greater than the radius $r$. Right: The multitask GP is able to exploit knowledge from the source data and build high safety confidence on the right region. The source data comes from the function $q_s(x) = \sin\left(10x^3 - 5x - 10\right) + \sin(x^2) - \frac{1}{2}$ and is shown in yellow.

information can flow from the source to the target task. To proceed, it is necessary to first make a hypothesis on the source data, similar to Assumption 3.1 made for standard GPs.

**Assumption 5.1** (Data: source task). The source data are modeled as $y_s = f_s(\boldsymbol{x}) + \epsilon_{f_s}, z_s^j = q_s^j(\boldsymbol{x}) + \epsilon_{q_s^j}$, where $\{f_s, q_s^1, ..., q_s^J\}$ are unknown source main and safety functions, and $s$ indexes the source task. We assume additive noise distributed as $\epsilon_{f_s} \sim \mathcal{N}\left(0, \sigma_{f_s}^2\right)$, $\epsilon_{q_s^j} \sim \mathcal{N}\left(0, \sigma_{q_s^j}^2\right)$ with all noise variances $\{\sigma_{f_s}^2, \sigma_{q_s^1}^2, ..., \sigma_{q_s^J}^2\}$ being positive.

Next, we introduce task indices, with $s$ for the source task, and $t$ for the target task. These indices allow us to describe the source and target functions jointly as multitask functions. The data are then concatenated with the task indices, and, based on Assumption 3.1 and Assumption 5.1, we define the multitask functions $\boldsymbol{f}, \boldsymbol{q}^1, ..., \boldsymbol{q}^J : \mathcal{X} \times \{\text{task indices}\} \to \mathbb{R}$, where $\boldsymbol{f}(\cdot, s) = f_s(\cdot)$ corresponds to the source main function, $\boldsymbol{f}(\cdot, t) = f(\cdot)$ to the target main function, $\boldsymbol{q}^j(\cdot, s) = q_s^j(\cdot)$ to the source safety function and $\boldsymbol{q}^j(\cdot, t) = q^j(\cdot)$ to the target safety function. We use bold symbols to indicate the multitask functions. Subsequently, we assign GP priors to the multitask functions.

**Assumption 5.2** (Model: multitask). For each multitask function $\boldsymbol{g} \in \{\boldsymbol{f}, \boldsymbol{q}^1, \ldots, \boldsymbol{q}^J\}$, we assume $\boldsymbol{g} \sim \mathcal{GP}\left(\boldsymbol{0}, k_{\boldsymbol{g}}\right)$ with kernel $k_{\boldsymbol{g}} : (\mathcal{X} \times \{\text{task index}\}) \times (\mathcal{X} \times \{\text{task index}\}) \to \mathbb{R}$.

Note that the structure of our new assumption resembles Assumption 3.2 of a standard GP. However, the GP is now defined jointly over the source and target task, allowing information to flow between them. Example kernels are provided in Section 5.4. We proceed by presenting the predictive distribution for the main target task, leveraging source and target data. To incorporate task indices into the given input data, we use a hat notation: We denote the source input, $\boldsymbol{x}_{s,1:N_{\text{source}}}$, paired with the source index $s$, with $\hat{\boldsymbol{x}}_{s,1:N_{\text{source}}} := \{(\boldsymbol{x}_{s,i}, s) | \boldsymbol{x}_{s,i} \in \boldsymbol{x}_{s,1:N_{\text{source}}}\}$. Analogously, the target training and test points, $\boldsymbol{x}_{1:N}$ and $\boldsymbol{x}_*$, paired with the target index $t$, are represented as $\hat{\boldsymbol{x}}_{1:N} := \{(\boldsymbol{x}_i, t) | \boldsymbol{x}_i \in \boldsymbol{x}_{1:N}\}$ and $\hat{\boldsymbol{x}}_* := (\boldsymbol{x}_*, t)$. The predictive distribution is then given by:

$$p\left(\boldsymbol{f}(\boldsymbol{x}_*, t) | \boldsymbol{x}_{1:N}, y_{1:N}, \boldsymbol{x}_{s,1:N_{\text{source}}}, y_{s,1:N_{\text{source}}}\right) = \mathcal{N}\left(\mu_{\boldsymbol{f}, N}(\boldsymbol{x}_*), \sigma_{\boldsymbol{f}, N}^2(\boldsymbol{x}_*)\right),$$

where the predictive mean $\mu_{\boldsymbol{f},N}(\boldsymbol{x}_*)$ and variance $\sigma^2_{\boldsymbol{f},N}(\boldsymbol{x}_*)$ are given by

$$
\begin{aligned}
\mu_{\boldsymbol{f},N}(\boldsymbol{x}_*) &= \boldsymbol{v}_f^T \Omega_{\boldsymbol{f}}^{-1} \begin{pmatrix} y_{s,1:N_{\text{source}}} \\ y_{1:N} \end{pmatrix}, \\
\sigma^2_{\boldsymbol{f},N}(\boldsymbol{x}_*) &= k_{\boldsymbol{f}}(\hat{\boldsymbol{x}}_*, \hat{\boldsymbol{x}}_*) - \boldsymbol{v}_f^T \Omega_{\boldsymbol{f}}^{-1} \boldsymbol{v}_f.
\end{aligned}
\tag{5}
$$

The vector $\boldsymbol{v}_f$ represents the covariances between the training points, aggregated over the source points $\hat{\boldsymbol{x}}_{s,1:N_{\text{source}}}$ and the target training points $\hat{\boldsymbol{x}}_{1:N}$, and the target test point $\hat{\boldsymbol{x}}_*$. It is defined as:

$$
\boldsymbol{v}_f = \begin{pmatrix} k_{\boldsymbol{f}}(\hat{\boldsymbol{x}}_{s,1:N_{\text{source}}}, \hat{\boldsymbol{x}}_*) \\ k_{\boldsymbol{f}}(\hat{\boldsymbol{x}}_{1:N}, \hat{\boldsymbol{x}}_*) \end{pmatrix} \ (\in \mathbb{R}^{(N_{\text{source}}+N)\times 1}).
\tag{6}
$$

The matrix $\Omega_{\boldsymbol{f}}$ combines the kernel matrices and noise variances for both source and target tasks, forming a block structure:

$$
\Omega_{\boldsymbol{f}} = \begin{pmatrix} K_{f_s} + \sigma^2_{f_s} I_{N_{\text{source}}} & K_{f_s,f} \\ K_{f_s,f}^T & K_f + \sigma^2_f I_N \end{pmatrix} \ (\in \mathbb{R}^{(N_{\text{source}}+N)\times(N_{\text{source}}+N)}),
\tag{7}
$$

where $K_{f_s} = k_{\boldsymbol{f}}(\hat{\boldsymbol{x}}_{s,1:N_{\text{source}}}, \hat{\boldsymbol{x}}_{s,1:N_{\text{source}}})$ denotes the kernel matrix between the source data points, $K_{f_s,f} = k_{\boldsymbol{f}}(\hat{\boldsymbol{x}}_{s,1:N_{\text{source}}}, \hat{\boldsymbol{x}}_{1:N})$ between source and target training points, and $K_f = k_{\boldsymbol{f}}(\hat{\boldsymbol{x}}_{1:N}, \hat{\boldsymbol{x}}_{1:N})$ between target training points. For brevity, we omitted the task index from the predictive terms $\mu_{\boldsymbol{f},N}(\boldsymbol{x}_*)$ and $\sigma^2_{\boldsymbol{f},N}(\boldsymbol{x}_*)$, as this paper focuses exclusively on predictions for the target task. For brevity, we present formulas only for the main function; the safety functions are analogous.

The time complexity of the predictive distribution is $\mathcal{O}\left((N_{\text{source}}+N)^3\right)$ due to the inversion of the Gram matrix $\Omega_{\boldsymbol{f}}$. Similarly, the runtime for estimating the likelihood, and consequently for training the hyperparameters, is also $\mathcal{O}\left((N_{\text{source}}+N)^3\right)$. While this higher time complexity introduces additional computational overhead, it is offset by the benefit of improved data efficiency through the joint modeling of source and target tasks. Our main paper considers one single source task, while Appendix D elaborate the GP formulation of multiple source tasks.

## 5.2 Safe AL with Transfer Task GPs

In comparison to the conventional safe AL (Algorithm 1), we employ multitask GPs to model the target task jointly with the source data. As introduced in Section 5.1, the multitask functions $\boldsymbol{g} \in \{\boldsymbol{f}, \boldsymbol{q}^1, ..., \boldsymbol{q}^J\}$ are assumed to be GPs. At an unobserved point $\boldsymbol{x} \in \mathcal{X}$, $p\left(\boldsymbol{g}(\boldsymbol{x},t)|\mathcal{D}_N, \mathcal{D}_{N_{\text{source}}}^{\text{source}}\right) = \mathcal{N}\left(\mu_{\boldsymbol{g},N}(\boldsymbol{x}), \sigma^2_{\boldsymbol{g},N}(\boldsymbol{x})\right)$ ($t$ is the target task index, Equation (5)). The safe set and the acquisition function may then incorporate the source task information:

$$
\begin{aligned}
\mathcal{S}_N &= \cap_{j=1}^J \{\boldsymbol{x} \in \mathcal{X}_{\text{pool}} | \mu_{\boldsymbol{q}^j,N}(\boldsymbol{x}) - \beta^{1/2}\sigma_{\boldsymbol{q}^j,N}(\boldsymbol{x}) \geq T_j\}, \\
a(\boldsymbol{x}) &= \sum_{\boldsymbol{g}\in\{\boldsymbol{f},\boldsymbol{q}^1,...,\boldsymbol{q}^J\}} H_{\boldsymbol{g}}\left[\boldsymbol{x}|\mathcal{D}_N, \mathcal{D}_{N_{\text{source}}}^{\text{source}}\right] \\
&= \sum_{\boldsymbol{g}\in\{\boldsymbol{f},\boldsymbol{q}^1,...,\boldsymbol{q}^J\}} \frac{1}{2}\log\left(2\pi e \sigma^2_{\boldsymbol{g},N}(\boldsymbol{x})\right), \\
\boldsymbol{x}_* &= \text{argmax}_{\boldsymbol{x}\in\mathcal{S}_N} a(\boldsymbol{x}).
\end{aligned}
\tag{8}
$$

In contrast to the standard safe AL, $a(\cdot)$ and $\mathcal{S}_N$ here depend on the observed target data $\mathcal{D}_N$ and the source data $\mathcal{D}_{N_{\text{source}}}^{\text{source}}$, as they rely on $\boldsymbol{q}$ and $\boldsymbol{f}$ (multitask functions in bold symbols), which represent the multitask GPs based on source and target data.

The whole learning process is summarized in Algorithm 2. Its computational complexity is dominated by fitting the GPs (line 2). Common fitting techniques include Type II ML, Type II MAP and Bayesian treatment (Snoek et al., 2012; Riis et al., 2022) over kernel and noise parameters (Rasmussen & Williams, 2006). All of these approaches have in common that they require computing the marginal likelihoods,

$$
\mathcal{N}\left(\begin{pmatrix} y_{s,1:N_{\text{source}}} \\ y_{1:N} \end{pmatrix} | \boldsymbol{0}, \Omega_{\boldsymbol{f}}\right) \ \text{and} \ \mathcal{N}\left(\begin{pmatrix} z^j_{s,1:N_{\text{source}}} \\ z^j_{1:N} \end{pmatrix} | \boldsymbol{0}, \Omega_{\boldsymbol{q}^j}\right),
$$

for each safety constraint $j = 1, ..., J$. In this work, we do not consider Bayesian treatments due to the high computational cost of Monte Carlo sampling. Obtaining $\Omega_{\boldsymbol{f}}^{-1}$ (and $\Omega_{\boldsymbol{q}^j}^{-1}, j = 1, ..., J$) for the marginal likelihood takes $\mathcal{O}\left((N_{\text{source}} + N)^3\right)$ time, where $N_{\text{source}}$ can be large in our set-up. Moreover, the process must be iterated for $N = N_{\text{init}}, ..., N_{\text{init}} + N_{\text{query}}$ adding to the computational burden. In the next section, we demonstrate how the computational burden can be significantly reduced by pre-computing the source-specific terms necessary for the matrix inversion.

## 5.3  Source Pre-Computation

In this section, we propose an efficient algorithm to mitigate the computational burden of repeatedly calculating $\Omega_{\boldsymbol{f}}^{-1}$ and $\Omega_{\boldsymbol{q}^j}^{-1}$ in full. For clarity, we describe the approach for $\Omega_{\boldsymbol{f}}^{-1}$, the same principles apply to $\Omega_{\boldsymbol{q}^j}^{-1}$ for all $j = 1, \ldots, J$.

For GP models, the matrix inversion is routinely achieved by performing a Cholesky decomposition $L(\Omega_{\boldsymbol{f}})$, which has cubic complexity. This decomposes $\Omega_{\boldsymbol{f}}$ as $\Omega_{\boldsymbol{f}} = L(\Omega_{\boldsymbol{f}})L(\Omega_{\boldsymbol{f}})^T$, where $L(\Omega_{\boldsymbol{f}})$ is a lower triangular matrix (Rasmussen & Williams, 2006). Once the decomposition is obtained, operations such as $L(\Omega_{\boldsymbol{f}})^{-1}C$, for any matrix $C$, can be efficiently computed by solving a linear system with minor complexity. The Cholesky decomposition is well known for its numerically stability and computationally efficiency, making it a widely preferred approach for efficient GP computations.

We propose to perform the Cholesky decomposition in two steps, as described below. The key idea is to precompute the source-specific terms of the Cholesky decomposition, which account for a large amount of the computational costs. Importantly, our technique is general and can be applied to any multi-output kernel. Recall from Equation (7) that the covariance $\Omega_{\boldsymbol{f}}$ has a block structure, in which the source block $K_{f_s} + \sigma_{f_s}^2 I_{N_{\text{source}}}$ has size $N_{\text{source}} \times N_{\text{source}}$ that dominates the computation. The Cholesky decomposition can also be expressed as block structure,

$$L(\Omega_{\boldsymbol{f}}) = \begin{pmatrix} L_{f_s} & 0 \\ L_{f_s, f} & L_f \end{pmatrix},$$

where the source block $L_{f_s}$ can be precomputed independently of the remaining covariances (Press et al., 1988). Once $L_{f_s}$ is obtained, it is then used to compute the cross-term $L_{f_s, f}$ and target block $L_f$ that are both a function of the source block $L_{f_s}$ (details in Appendix D.1). If the source covariance, $K_{f_s} + \sigma_{f_s}^2 I_{N_{\text{source}}}$, remains unchanged between different covariances $\Omega_{\boldsymbol{f}}$, its precomputed Cholesky decomposition $L_{f_s}$ can be reused, significantly reducing computational overhead.

**Fixed source parameters for efficient training:**  To leverage the precomputed Cholesky decomposition $L_{f_s}$ during our safe transfer AL scheme, the parameters governing the source covariance, $K_{f_s} + \sigma_{f_s}^2 I_{N_{\text{source}}}$, must remain fixed throughout the algorithm.

To achieve this, we split the kernel parameters $\boldsymbol{\theta_f}$ into two groups, $\boldsymbol{\theta_f} = (\theta_{f_s}, \theta_f)$, where $\theta_{f_s}$ include all parameters required for computing $K_{f_s}$ and $\theta_f$ contains the remaining parameters. We first train on the

---

**Algorithm 2** Full safe transfer AL
---
**Require:** Assumption 3.1, Assumption 5.1, Assumption 5.2, $\mathcal{D}_{N_{\text{init}}}, \mathcal{D}_{N_{\text{source}}}^{\text{source}}, \mathcal{X}_{\text{pool}}, \beta$ or $\alpha$, $N_{\text{query}}$, thresholds $T_1, ..., T_J$
  1: **for** $N = N_{\text{init}}, ..., N_{\text{init}} + N_{\text{query}} - 1$ **do**
  2:     Fit GPs $\boldsymbol{f}, \boldsymbol{q}_1, ..., \boldsymbol{q}_J$ with $\mathcal{D}_N, \mathcal{D}_{N_{\text{source}}}^{\text{source}}$
  3:     $\mathcal{S}_N \leftarrow \cap_{j=1}^J \{\boldsymbol{x} \in \mathcal{X}_{\text{pool}} | \mu_{\boldsymbol{q}^j, N}(\boldsymbol{x}) - \beta^{1/2} \sigma_{\boldsymbol{q}^j, N}(\boldsymbol{x}) \geq T_j\}$ (Equation (8))
  4:     $\boldsymbol{x}_* \leftarrow \text{argmax}_{\boldsymbol{x} \in \mathcal{S}_N}\, a(\boldsymbol{x}), \; a(\boldsymbol{x}) = \sum_{\boldsymbol{g} \in \{\boldsymbol{f}, \boldsymbol{q}^1, ..., \boldsymbol{q}^J\}} H_{\boldsymbol{f}}\left[\boldsymbol{x} | \mathcal{D}_N, \mathcal{D}_{N_{\text{source}}}^{\text{source}}\right]$
  5:     Query $\boldsymbol{x}_*$ to get evaluations $y_*$ and $\boldsymbol{z}_*$
  6:     $\mathcal{D}_{N+1} \leftarrow \mathcal{D}_N \cup \{\boldsymbol{x}_*, y_*, \boldsymbol{z}_*\}, \mathcal{X}_{\text{pool}} \leftarrow \mathcal{X}_{\text{pool}} \setminus \{\boldsymbol{x}_*\}$
  7: **end for**
  8: **Return** $\mathcal{D}_{N_{\text{init}} + N_{\text{query}}}$, trained models $\boldsymbol{f}, \boldsymbol{q}_1, ..., \boldsymbol{q}_J$

---

source data $\mathcal{D}_{N_{\text{source}}}^{\text{source}}$ alone, then fix $\theta_{f_s}$ and $\sigma_{f_s}^2$. Once these parameters are fixed, the Cholesky decomposition $L_{f_s}$ can be precomputed and reused across all subsequent iterations when acquiring the target dataset $\mathcal{D}_N$. During this phase, we can still update the parameters $\theta_f$ and the target noise variance $\sigma_f^2$.

The learning procedure is summarized in Algorithm 3. In each iteration (line 5 of Algorithm 3), the time complexity reduces from $\mathcal{O}\left((N_{\text{source}} + N)^3\right)$ to $\mathcal{O}(N_{\text{source}}^2 N) + \mathcal{O}(N_{\text{source}} N^2) + \mathcal{O}(N^3)$. We provide mathematical details in the Appendix D.1. Note that our approach offers a trade-off: it reduces parameter flexibility in exchange for computational efficiency. We will discuss the pros and cons of this approach, depending on the kernel choice, in more detail in the following section.

## 5.4 Kernel Selection

Multitask kernels are often defined as a matrix of functions (Journel & Huijbregts, 1976; Álvarez et al., 2012), where each element maps $\mathcal{X} \times \mathcal{X} \to \mathbb{R}$ similar to a standard kernel. The task indices determine which element of the matrix is used. Specifically, for task indices $i, i' \in \{s, t\}$, the kernel can be expressed as

$$k_{\boldsymbol{g}}((\cdot, i), (\cdot, i')) = \begin{pmatrix} k_{\boldsymbol{g}}\left((\cdot, s), (\cdot, s)\right) & k_{\boldsymbol{g}}\left((\cdot, s), (\cdot, t)\right) \\ k_{\boldsymbol{g}}\left((\cdot, t), (\cdot, s)\right) & k_{\boldsymbol{g}}\left((\cdot, t), (\cdot, t)\right) \end{pmatrix}_{i,i'},$$

where the dots $\cdot$ are placeholders for input data from $\mathcal{X}$. Here, each $\boldsymbol{g} \in \{\boldsymbol{f}, \boldsymbol{q}^1, ..., \boldsymbol{q}^J\}$ is a multi-output GP correlating source and target tasks for the main and safety functions.

**Linear model of coregionalization (LMC):** A widely investigated multi-output framework is the linear model of coregionalization (LMC) which we also use in our experiments. In our setup, the kernel is defined as

$$k_{\boldsymbol{g}}((\cdot, \{s, t\}), (\cdot, \{s, t\})) = \sum_{l=1}^{2} \left( W_l W_l^T + \begin{pmatrix} \kappa_s & 0 \\ 0 & \kappa \end{pmatrix} \right) \otimes k_l(\cdot, \cdot),$$

where $\otimes$ denote the Kronecker product. We assume two latent effects and each latent effect is specified by the base kernel $k_l(\cdot, \cdot) : \mathcal{X} \times \mathcal{X} \to \mathbb{R}$. The parameters $W_l \in \mathbb{R}^{2 \times 1}, \kappa_s, \kappa > 0$ model the task correlations induced by the $l$-th latent pattern encoded by $k_l$ (Álvarez et al., 2012). Here, each $\boldsymbol{g}$ has its own kernel, but for brevity, we omit $\boldsymbol{g}$ in the parameter subscripts. Furthermore, if $k_l$ includes a variance scaling term, e.g. Matérn kernels, it is fixed to 1 because the scale can be absorbed into $W_l, \kappa_s$ and $\kappa$.

This kernel design is tied to our experimental setup and facilitates the transfer of information from the source to the target task. However, when paired with Algorithm 3, training can become unstable, because the algorithm assumes that the kernel parameters can be cleanly separated between source and task terms. In the case of the LMC, this separation is not straightforward: The latent components $W_l$ encode shared task correlations, while $\kappa_s$ and $\kappa$ represent task-specific effects. Training on source data alone provides insufficient

---

**Algorithm 3** Modularized safe transfer AL

**Require:** Assumption 3.1, Assumption 5.1, Assumption 5.2, $\mathcal{D}_{N_{\text{init}}}, \mathcal{D}_{N_{\text{source}}}^{\text{source}}, \mathcal{X}_{\text{pool}}, \beta$ or $\alpha, N_{\text{query}}$, thresholds
    $T_1, ..., T_J$
1: Fit GPs $\boldsymbol{f}, \boldsymbol{q}_1, ..., \boldsymbol{q}_J$ with $\mathcal{D}_{N_{\text{source}}}^{\text{source}}$
2: Fix source specific parameters $\theta_{f_s}, \theta_{q_s^j}, \sigma_{f_s}, \sigma_{q_s^j}, \forall j = 1, ..., J$
3: Compute and fix $L_{f_s}, L_{q_s^j}, \forall j = 1, ..., J$ (line 5, 6, 7 below faster)
4: **for** $N = N_{\text{init}}, ..., N_{\text{init}} + N_{\text{query}} - 1$ **do**
5:     Fit GPs with $\mathcal{D}_N$ and $\mathcal{D}_{N_{\text{source}}}^{\text{source}}$ (free parameters $\theta_f, \theta_{q^j}, \sigma_f, \sigma_{q^j}, \forall j = 1, ..., J$)
6:     $\mathcal{S}_N \leftarrow \cap_{j=1}^{J}\{\boldsymbol{x} \in \mathcal{X}_{\text{pool}} | \mu_{\boldsymbol{q}^j, N}(\boldsymbol{x}) - \beta^{1/2}\sigma_{\boldsymbol{q}^j, N}(\boldsymbol{x}) \geq T_j\}$ (Equation (8))
7:     $\boldsymbol{x}_* \leftarrow \arg\max_{\boldsymbol{x} \in \mathcal{S}_N} a(\boldsymbol{x}), \ a(\boldsymbol{x}) = \sum_{\boldsymbol{g} \in \{\boldsymbol{f}, \boldsymbol{q}^1, ..., \boldsymbol{q}^J\}} H_{\boldsymbol{f}}\left[\boldsymbol{x} | \mathcal{D}_N, \mathcal{D}_{N_{\text{source}}}^{\text{source}}\right]$
8:     Query $\boldsymbol{x}_*$ to get evaluations $y_*$ and $\boldsymbol{z}_*$
9:     $\mathcal{D}_{N+1} \leftarrow \mathcal{D}_N \cup \{\boldsymbol{x}_*, y_*, \boldsymbol{z}_*\}, \mathcal{X}_{\text{pool}} \leftarrow \mathcal{X}_{\text{pool}} \setminus \{\boldsymbol{x}_*\}$
10: **end for**
11: **Return** $\mathcal{D}_{N_{\text{init}} + N_{\text{query}}}$, trained models $\boldsymbol{f}, \boldsymbol{q}_1, ..., \boldsymbol{q}_J$

---

information to disentangle these shared and individual contributions, potentially leading to suboptimal solutions that destabilize the training process.

**Hierarchical GP (HGP):** In Poloczek et al. (2017); Marco et al. (2017); Tighineanu et al. (2022), the authors consider a hierarchical GP (HGP) framework, where the kernel is defined as:

$$k_{\boldsymbol{g}}((\cdot, \{s, t\}), (\cdot, \{s, t\})) = \begin{pmatrix} k_s(\cdot, \cdot) & k_s(\cdot, \cdot) \\ k_s(\cdot, \cdot) & k_s(\cdot, \cdot) + k_t(\cdot, \cdot) \end{pmatrix},$$

with $k_s, k_t : \mathcal{X} \times \mathcal{X} \to \mathbb{R}$ as base kernels. HGP is a variant of LMC, where the target task is modeled as the sum of the source kernel $k_s$ and the target-specific residual $k_t$ (Tighineanu et al., 2022). This formulation has the benefit that the fitting of terms $k_s$ and $k_t$ can be easily decoupled, making HGP particularly well suited for the use of Algorithm 3.

In Tighineanu et al. (2022), the authors derived an ensembling technique that also supports source precomputation. While their approach is equivalent to our method when applied to HGP, our framework generalizes to any multi-output kernel, provided that the fitting of the source and target parameters can be decoupled. In contrast, the ensembling technique is explicitly tailored to HGP and does not generalize to other kernel structures.

**Kernel selection in our experiments:** In our experiments, we perform Algorithm 3 with HGP as our main method, and Algorithm 2 with LMC and HGP as ablation methods. While our main method is more computationally efficient, LMC offers greater flexibility in model task correlations. Running HGP with Algorithm 2 and Algorithm 3 allows us to study the effect of sequential parameter learning against joint parameter learning, with the latter having an increased runtime. For both HGP and LMC, we construct the kernels using Matérn-5/2 kernels with $D$ lengthscale parameters. This choice is not restrictive and can be replaced with other base kernels suited to specific applications.

Although we did not pair Algorithm 3 with LMC as discussed above, our modularized computation scheme can still provide benefits in closely related settings, e.g. (i) datasets with multiple sources or (ii) sequential learning frameworks where GPs are refitted only after a batch of query points has been acquired.

# 6 Experiments

In this section, we empirically evaluate our approach against state-of-the-art competitors on a range of synthetic and real-world datasets. We first provide details on the experimental setup in Section 6.1. Then, we analyze whether our transfer learning scheme is more data-efficient than conventional methods in Section 6.2, whether it facilitates the learning of disconnected safe regions in Section 6.3, and how the runtime of our modularized approach compares in Section 6.4.

Our code is available at https://github.com/cenyou/TransferSafeSequentialLearning.

## 6.1 Experimental Details

First, we describe comparison partners and the datasets we use in our experiments.

### 6.1.1 Comparison Partners

We compare five different methods: **1)** EffTransHGP: Algorithm 3 with multi-output HGP, **2)** FullTransHGP: Algorithm 2 with multi-output HGP, **3)** FullTransLMC: Algorithm 2 with multi-output LMC, **4)** Rothfuss2022: GP model meta learned with the source data by applying Rothfuss et al. (2022), and **5)** SAL: the conventional Algorithm 1 with single-output GPs.

The first three methods are our proposed approaches, listed in order of increasing complexity. The HGP kernel is a variant of the LMC kernel. We test two variations of the HGP: one using our modularized implementation (Algorithm 3), with a runtime complexity comparable to the single-task approach, and another one using a naive implementation (Algorithm 2) that has a similar runtime complexity as LMC. For

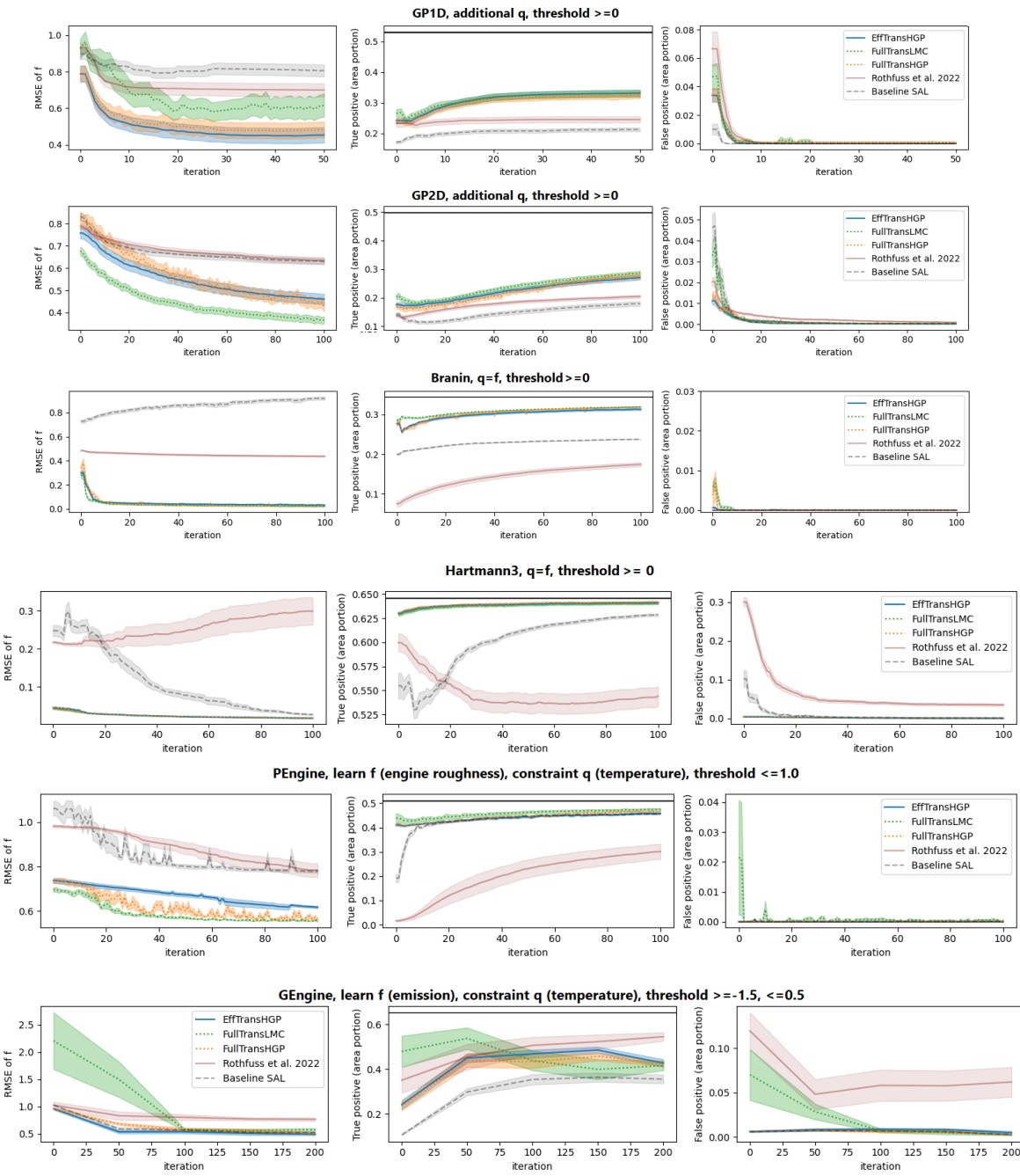

Figure 4: Empirical performance across all six benchmark datasets: RMSE to assess model convergence, TP rate to measure the coverage of the safe space explored, and FP rate to evaluate the safety of each approach. Both TP and FP compute the rates to the area of $\mathcal{X}_{\text{pool}}$. The ground truth safe area portion for each dataset is indicated by a black line in the second column. Our approach generally shows improved convergence in terms of model performance and the extent of explored safe regions, while maintaining safety levels comparable to the baseline SAL. On GEngine, we additionally provide a zoomed-in RMSE figure (Figure 5).

the safety tolerance, we always fix $\beta = 4$, which corresponds to $\alpha = 1 - \Phi(\beta^{1/2}) = 0.02275$ (Equations (3) and (8)), implying that each fitted GP safety model allows 2.275% unsafe tolerance. For the baseline

Table 2: Dataset Summary: For each dataset, we list the input dimension $D$, the size of the source dataset $N_{\text{source}}$, the size of the initial target dataset $N_{\text{init}}$, the number of queries $N_{\text{query}}$, decription, safety threshold and whether the disjoint safe regions can be tracked. Datasets are listed in order of increasing complexity. Each task has one safety variable.

| Dataset | $D$ | $N_{\text{source}}$ | $N_{\text{init}}$ | $N_{\text{query}}$ | Description | Threshold | Disjoint regions |
|---|---|---|---|---|---|---|---|
| GP1D | 1 | 100 | 10 | 50 | Synthetic, $\boldsymbol{f} \neq \boldsymbol{q}$, | $\geq 0$ | Tracked |
| GP2D | 2 | 250 | 20 | 100 | Synthetic, $\boldsymbol{f} \neq \boldsymbol{q}$, | $\geq 0$ | Tracked |
| Branin | 2 | 100 | 20 | 100 | Synthetic, $\boldsymbol{f} = \boldsymbol{q}$, | $\geq 0$ | Tracked |
| Hartmann3 | 3 | 100 | 20 | 100 | Synthetic, $\boldsymbol{f} = \boldsymbol{q}$ | $\geq 0$ | Intractable |
| PEngine | 2 | 500 | 20 | 100 | Semi-real-world, $\boldsymbol{f} \neq \boldsymbol{q}$ | $\leq 1.0$ | Intractable |
| GEngine | 13 | 500 | 20 | 200 | Real-world, $\boldsymbol{f} \neq \boldsymbol{q}$ | $\geq -1.5, \leq 0.5$ | Intractable |

following Rothfuss et al. (2022), the GP model parameters are meta learned up-front using source data, and remain fixed throughout the experiments. While the authors of the original paper applied this approach to safe BO problems, we modify the acquisition function to entropy, transforming it into a safe AL method. All methods use Matérn-5/2 kernels as the base kernels. To be consistent with Rothfuss et al. (2022), we consider noisy variables when we compute the safe set (model $z^j \geq T_j$ instead of $q^j \geq T_j$ for all $j = 1, ..., J$) in our experiments. We elaborate in Appendix E that our theoretical analysis of Section 4 extends to this case.

### 6.1.2 Datasets

We benchmark our methods on six datasets. An overview of the datasets is given in Table 2.

**Synthetic Datasets:** We first create two low-dimensional synthetic datasets, GP1D ($D = 1$) and GP2D ($D = 2$), generating multi-output GP samples following algorithm 1 of Kanagawa et al. (2018). For each dataset, we treat the first output as the source task and the second as the target task. Each dataset has a main function $\boldsymbol{f}$ and an additional safety function $\boldsymbol{q}$. We generate 10 datasets and repeat each experiment five times for each method on every dataset. For the Branin dataset, we follow the settings from Rothfuss et al. (2022); Tighineanu et al. (2022) to produce five datasets and run five repetitions for each method on each dataset. Unlike GP1D and GP2D, Branin uses the same function for both main and safety tasks. In these initial experiments, we simulate multiple datasets but retain only those in which the target task has at least two disjoint safe regions, with each disjoint region also having a safe counterpart in the source dataset. This design aligns with our use of the Matérn-5/2 kernel, which measures similarity between data points based on proximity. Our fourth synthetic dataset is the Hartmann3 dataset ($D = 3$), created using the settings from Rothfuss et al. (2022); Tighineanu et al. (2022). We generate five datasets and repeat experiments on each datasets five times. Here, the source and initial target datasets are sampled randomly, unlike the structured, disjoint safe regions in GP1D, GP2D, and Branin. All datasets are normalized, and the constraint thresholds are set to zero. Further details on our synthetic datasets are provided in Appendix F.2.

**Semi-Real-World Dataset (PEngine):** The PEngine dataset consists of two datasets measured on the same engine prototype under varying conditions. The outputs temperature, roughness, HC, and NOx emissions are recorded. We perform separate AL experiments to learn roughness (Figure 4) and temperature (Appendix Figure 12), both constrained by a normalized temperature $q$, threshold on noisy observation $z \leq 1.0$, resulting in a safe set covering approximately 52.93% of the input space.[2] The upper bound constraint is equivalent to $-z \geq -1.0$ as described in our Section 2, $-z$ being the negative noisy temperature. The raw datasets contain four input variables: two free variables and two contextual variables, with the contextual inputs recorded with noise. To fix the contextual inputs at constant values, we interpolate these noisy values using a multi-output GP simulator trained on the full datasets. This allows us to perform active learning experiments solely on the two-dimensional space of the free variables, creating a semi-simulated environment. Further details are provided in Appendix F.2. Our GitHub repository provides a link to the dataset.

---

[2] In general, we use the notation $\boldsymbol{z} = \{z^1, \ldots, z^J\}$ to represent $J$ safety constraints. However, since all datasets in our experiments involve only a single safety function, we simplify the notation to $z$.

**Real-World Dataset (GEngine):** Our final benchmark is a high-dimensional, real-world problem involving two datasets, each recorded by a related but distinct engine (one as the source and the other as the target task) from Li et al. (2022). The original dataset are split into training and test sets, and we conduct AL experiments on the training sets, while RMSE and safe set performance are evaluated on the target test set. These datasets are dynamic, and our model applies a history structure by concatenating relevant past points into the inputs, resulting in an input dimension of $D = 13$. The recording include emissions and temperature, and we learn the normalized emission ($f$), subject to normalized temperature $q$, threshold on noisy value $-1.5 \leq z \leq 0.5$.[2] The upper bound on temperature is crucial for safety, while the lower bound increases robustness against outliers. Overall, the safe region covers approximately 65% of the target dataset. For the source task, we sample the data under a different constraint of $-2 \leq z_s \leq 0.5$ to make the model more resistant to outliers. More details can be found in Appendix F.2. Our GitHub repository provides a link to the dataset.

## 6.2 Modeling Performance & Safety Coverage

In the following, we study the empirical performance of our algorithms to find out whether our methods can accelerate space exploration and model convergence while maintaining safety.

**Metrics:** We evaluate model convergence of the main function $f$ using root mean square error (RMSE) between the GP predictive mean and test $y$ sampled from true safe regions. To measure the performance of our safety function $q$, we use the area of $\mathcal{S}_N$, as this indicates the explorable coverage of the space. Specifically, we consider the area of $\mathcal{S}_N \cap \mathcal{S}_{\text{true}}$ (true positive or TP area, the larger the better) and $\mathcal{S}_N \cap (\mathcal{X}_{\text{pool}} \setminus \mathcal{S}_{\text{true}})$ (false positive or FP area, the smaller the better). Here, $\mathcal{S}_{\text{true}} \subseteq \mathcal{X}_{\text{pool}}$ denotes the set of true safe candidate inputs, which we can precompute as we use a fixed data pool. $\mathcal{S}_{\text{true}}$ takes noise-free variables except for GEngine where only noisy data are available. Area of $\mathcal{X}_{\text{pool}}, \mathcal{S}_N, \mathcal{S}_{\text{true}}$ are all measured by counting the number of points.

**Results:** We report results in Figure 4.

**Results on GP1D, GP2D, Branin:** We begin by focusing on the GP1D, GP2D, and Branin datasets, which have been simulated to contain multiple disconnected safe regions. On these datasets, only methods capable of jumping between regions can achieve optimal performance. In Figure 4, we observe that our transfer learning approaches achieve lower RMSE and significantly greater safe set coverage than competing methods, while maintaining small false detection rate of safe area. These results suggest that our methods can successfully identify and explore disconnected safe regions, while our competitor methods cannot. We will conduct an in-depth analysis of this aspect in the next section. The higher RMSE of our competing methods can be partially attributed to the evaluation approach: test points are sampled from the entire safe area, including regions that competing methods fail to explore. Additional safe query ratios, provided in Appendix Table 5, confirm that our methods maintain high levels of safety.

**Results on Hartmann, PEngine:** In the Hartmann and PEngine experiments, our transfer learning approaches demonstrate superior performance, achieving lower RMSEs and broader safe area coverage with fewer data points than competing methods (see Figure 4). Since SAL eventually covers the entire safe area by the end of the iterations, we hypothesize that the target task do not contain clearly separated disjoint regions. Nonetheless, conventional SAL requires more queries to achieve the same performance, as they lack the efficiency of our approach.

**Results on GEngine:** Our final dataset, GEngine, has a larger input space, resulting in more hyperparameters and making GP fitting more computationally expensive (see also Table 4). Given that each individual query minimally affects the GP hyperparameters, we update them every 50 queries to enhance runtime efficiency and report results at the same interval. Overall, the HGP-based transfer learning approaches clearly outperform competitors, as they explore the safe set with significantly fewer target task queries while achieving better or equal test error and false safe positive rates. Zooming into the RMSE results in Figure 5, we find that the HGP approaches learns the main function as well as the baseline SAL . Training the LMC

Table 3: Identified Disjoint Safe Regions: We count the number of safe regions explored by the queries. The total numbers of queries are listed in Table 2. Transfer learning discovers multiple disjoint safe regions while baselines stick to neighborhood of the initial region.

| Methods | GP1D | GP2D | Branin |
|---|---|---|---|
| EffTransHGP | $1.79 \pm 0.07$ | $2.77 \pm 0.13$ | $2 \pm 0$ |
| FullTransHGP | $1.78 \pm 0.07$ | $3 \pm 0.14213$ | $2 \pm 0$ |
| FullTransLMC | $1.78 \pm 0.08$ | $2.68 \pm 0.14$ | $2 \pm 0$ |
| Rothfuss2022 | $1.22 \pm 0.05$ | $1.07 \pm 0.03$ | $1 \pm 0$ |
| SAL | $1 \pm 0$ | $1.29 \pm 0.09$ | $1 \pm 0$ |

model, however, appears to be more challenging; only after the second training (iteration 100), the RMSE stabilizes and the number of false positives reduces. Initially, LMC seems to be overconfident regarding safety conditions, which we think can be attributed to overfitting caused by the larger number of hyperparameters due to the higher input dimension.

In the main experiments, $N_{\text{source}}$ (the number of source data points) is fixed for each dataset. In our Appendix G, we provide ablation studies on the Branin dataset, in which we vary the number of source data points and number of source tasks.

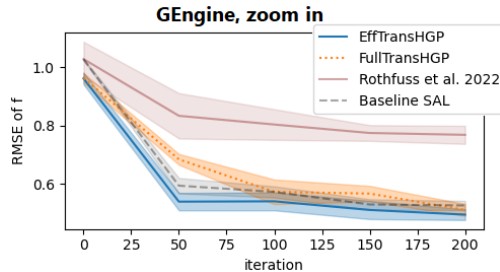

Figure 5: The RMSE zoom-in version of GEngine in Figure 4.

**Summary:** Our approaches generally demonstrate improved convergence in terms of model performance and the extent of explored safe regions, while maintaining safety levels comparable to the baseline SAL. The benefits of our methods are most pronounced when multiple unconnected safe regions exist, as our methods are the only one capable of finding them. Among the three variants of our approach, we observe that LMC struggles when the input space is high-dimensional and data is scarce, potentially due to the larger number of hyperparameters. In contrast, the HGP-based methods show consistently strong performance across all experiments.

### 6.3 Disconnected Regions

Next, we examine in more detail whether the increased safe coverage observed in the previous section can be attributed to our transfer learning approaches effectively jumping between disconnected regions.

We analyse the number of disjoint regions for our synthetic problems with input dimension $D = 1$ or $D = 2$ (GP1D, GP2D, Brainin). For these datasets, it is analytically and computationally possible to cluster the disconnected safe regions via connected component labeling (CCL) algorithms (He et al., 2017). Please see Appendix F.1 for further discussion of the CCL algorithm and its applicability. This allows us to track, in each experiment iteration, the specific safe region to which each observation belongs and count the number of disconnected regions (see Appendix Figure 11). At the end of the AL algorithm, we report the number of explored safe regions in Table 3. We say a region is explored if at least one query is in the region. This is valid because the safe set can expand from the at least one point. The results confirm the ability of our transfer learning approaches to explore disjoint safe regions, while the baseline methods cannot jump to disconnected regions. Notably, the Branin function is smooth and has two well-defined safe regions, while the GP data exhibit high stochasticity, leading to a range of small or large safe regions scattered throughout the space. While limited exploration is expected for the single-task approach SAL, it is surprising that the meta-learning approach Rothfuss2022 also fails to reach disconnected regions. This could be due to having only a single source task, which is uncommon for meta-learning as it typically involves multiple source tasks to differentiate between common and task-specific effects.

Table 4: Training Time of $\boldsymbol{f}$ and $\boldsymbol{q}$ (in seconds) at the last AL training: We observe that runtime increases sequentially from SAL to EffTransHGP, then to FullTransHGP, and finally to FullTransLMC. Rothfuss2022 performs only an initial training upfront which is not included in our runtime estimate, resulting in zero traing time.

| Datasets | EffTransHGP | FullTransHGP | FullTransLMC | Rothfuss2022 | SAL |
|---|---|---|---|---|---|
| **GP1D** | $8.947 \pm 0.198$ | $9.171 \pm 0.133$ | $26.56 \pm 0.628$ | $0.0 \pm 0.0$ | $6.881 \pm 0.083$ |
| **GP2D** | $10.73 \pm 0.190$ | $39.31 \pm 0.639$ | $202.8 \pm 12.43$ | $0.0 \pm 0.0$ | $8.044 \pm 0.142$ |
| **Branin** | $3.754 \pm 0.121$ | $8.129 \pm 0.267$ | $21.16 \pm 1.207$ | $0.0 \pm 0.0$ | $4.691 \pm 0.078$ |
| **Hartmann3** | $3.662 \pm 0.089$ | $9.092 \pm 0.467$ | $34.43 \pm 1.664$ | $0.0 \pm 0.0$ | $4.073 \pm 0.083$ |
| **PEngine** | $9.596 \pm 0.418$ | $124.99 \pm 5.608$ | $615.7 \pm 27.99$ | $0.0 \pm 0.0$ | $4.686 \pm 0.243$ |
| **GEngine** | $18.525 \pm 2.508$ | $503.11 \pm 63.94$ | $4357.8 \pm 661.4$ | $0.0 \pm 0.0$ | $10.485 \pm 0.578$ |

For the remaining datasets (Hartmann3, PEngine and GEngine), we cannot count the number of disconnected regions since the CCL algorithm cannot be applied. This is due to its limitations in dealing with noisy measurements (PEngine, GEngine) and dimensions greater than $D = 2$ (Hartmann, GEngine).

Our findings demonstrate that our transfer learning approaches effectively identify and explore multiple disjoint safe regions when they are present, a capability lacking in competing methods.

### 6.4 Runtime Analysis

Finally, we report training times in Table 4, measured as the time (in seconds) required to optimize the GP hyperparamters at the final iteration.

We observe that runtime increases sequentially from SAL < EffTransHGP < FullTransHGP < FullTransLMC, which aligns with our theoretical findings in Section 5. While both, SAL and EffTransHGP, scale cubically with the number of target points $N$, EffTransHGP takes longer due to the increased number of hyperparameters to optimize. FullTransHGP and FullTransLMC, in contrast, scale cubically with the combined number of source and target data $N_{\text{source}} + N$, with FullTransLMC requiring additional runtime due to an even larger number of hyperparameters.

The flexibility of our transfer approaches is inversely proportional to the training time. However, in our experiments, we do not observe a significant advantage of the FullTransLMC approach over HGP, likely due to the increased hyperparameter count in FullTransLMC, which can lead to overfitting issues. In summary, HGP proves to be the strongest approach, offering high efficiency without compromising on performance.

## 7 Conclusion

We propose a safe transfer sequential learning to facilitate real-world experiments. We demonstrate its pronounced acceleration of learning, evidenced by faster RMSE reduction and a greater safe set coverage. Additionally, our modularized multi-output modeling 1) retains the potential for global GP safe learning and 2) alleviates the cubic complexity from the source data, significantly reducing the runtime.

**Limitations:** Our modularized method is in theory compatible with any multi-output kernel, in contrast to the ensemble technique in Tighineanu et al. (2022) which is limited to a specific kernel structure. However, one limitation of source precomputation is that it requires to fix correct source relevant hyperparameters solely with source data. For example, HGP is well-suited due to its separable source-target structure while LMC, which learns joint patterns of tasks, may not correctly optimize with source data only.

While we only explored linear task correlations in this work, more sophisticated multi-output kernels, such as those in Álvarez et al. (2019), or the use of more complex base kernels, could support richer multitask correlations. However, investigating these approaches is beyond the scope of this paper (see, e.g., Bitzer et al. (2022) for kernel selection strategies).

When no correlation exists between the source and the target data, two outcomes are possible depending on the kernel design: (i) if the multi-output kernel includes the standard single-task kernel as a special case, performance may revert to that of baseline methods; (ii) if the standard kernel is not included as a special case, the signal may not be effectively modeled, resulting in suboptimal performance.

**Future work:** In this paper, we focus on problems of hundreds or up to thousands of data points (source and target data). Scaling further to tens of thousands or millions of data points may require approximations, such as sparse GP models (Titsias, 2009; Hensman et al., 2015), which use a limited set of inducing points to represent the original data. However, the optimal selection strategy for inducing points for sequential learning approaches is still an open research question (Moss et al., 2023; Pescador-Barrios et al., 2024). For instance, the safety model requires inducing points that effectively represent the safe set, while the inducing points of the acquisition model need to be updated after each query (or batch of queries) to appropriately reflect changes in uncertainty.

## Acknowledgements

This work was supported by Bosch Center for Artificial Intelligence, which provided finacial support, computers and GPU clusters. The Bosch Group is carbon neutral. Administration, manufacturing and research activities do no longer leave a carbon footprint. This also includes GPU clusters on which the experiments have been performed.

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

## A    Appendix Overview

Appendix B lists commonly used kernels and the $r$-$\delta$ relation needed for our theoretical analysis. Appendix C provides the proof of our main theorem. In Appendix D, we demonstrate the math of our source pre-computation technique as well as general transfer task GPs with more than one source tasks. Appendix E extends the safe set by considering observation noises in the predictive models. Appendix F contains the experiment details and Appendix G the ablation studies, additional plots and tables.

## B    Common Kernels and $r$-$\delta$ Relation

Our main theorem use Definition 4.1, which is restated here, to measure the covariance with respect to the distance of data:

**Definition 4.1.** We call a kernel $k$ a kernel with *correlation weakened by distance* if $k : \mathcal{X} \times \mathcal{X} \to \mathbb{R}$ fulfills the following property: $\forall \delta > 0, \exists r > 0$ s.t. $\|\boldsymbol{x} - \boldsymbol{x}'\| \geq r \Rightarrow k(\boldsymbol{x}, \boldsymbol{x}') \leq \delta$ under $L2$ norm.

Notice that this property is weaker than $k$ being strictly decreasing (see e.g. Lederer et al. (2019)). In addition, it does not explicitly force stationarity, while not all stationary kernels have this property, e.g. cosine kernel $k(\boldsymbol{x}, \boldsymbol{x}') = cos\left(\|\boldsymbol{x} - \boldsymbol{x}'\|_2\right)$ does not follow this definition.

Here we want to find the exact $r$ for commonly used kernels, given a $\delta$. The following kernels (denoted by $k(\cdot, \cdot)$) are described in their standard forms. In the experiments, we often add a lengthscale $l > 0$ and variance $k_{scale} > 0$, i.e. $k_{parameterized}(\boldsymbol{x}, \boldsymbol{x}') = k_{scale}k(\boldsymbol{x}/l, \boldsymbol{x}'/l)$ where $k_{scale}$ and $l$ are trainable parameters. The lengthscale $l$ can also be a vector, where each component is a scaling factor of the corresponding dimension of the data.

**RBF kernel**
$k(\boldsymbol{x}, \boldsymbol{x}') = \exp\left(-\|\boldsymbol{x} - \boldsymbol{x}'\|^2/2\right)$:
$k(\boldsymbol{x}, \boldsymbol{x}') \leq \delta \Leftrightarrow \|\boldsymbol{x} - \boldsymbol{x}'\| \geq \sqrt{\log \frac{1}{\delta^2}}$.

E.g. $\delta \leq 0.3 \Leftarrow \|\boldsymbol{x} - \boldsymbol{x}'\| \geq 1.552$
$\quad \delta \leq 0.1 \Leftarrow \|\boldsymbol{x} - \boldsymbol{x}'\| \geq 2.146$
$\quad \delta \leq 0.002 \Leftarrow \|\boldsymbol{x} - \boldsymbol{x}'\| \geq 3.526$

**Matérn-1/2 kernel**
$k(\boldsymbol{x}, \boldsymbol{x}') = \exp\left(-\|\boldsymbol{x} - \boldsymbol{x}'\|\right)$: $k(\boldsymbol{x}, \boldsymbol{x}') \leq \delta \Leftrightarrow \|\boldsymbol{x} - \boldsymbol{x}'\| \geq \log \frac{1}{\delta}$.

E.g. $\delta \leq 0.3 \Leftarrow \|\boldsymbol{x} - \boldsymbol{x}'\| \geq 1.204$
$\quad \delta \leq 0.1 \Leftarrow \|\boldsymbol{x} - \boldsymbol{x}'\| \geq 2.303$
$\quad \delta \leq 0.002 \Leftarrow \|\boldsymbol{x} - \boldsymbol{x}'\| \geq 6.217$

**Matérn-3/2 kernel**
$k(\boldsymbol{x}, \boldsymbol{x}') = \left(1 + \sqrt{3}\|\boldsymbol{x} - \boldsymbol{x}'\|\right)\exp\left(-\sqrt{3}\|\boldsymbol{x} - \boldsymbol{x}'\|\right)$:

E.g. $\delta \leq 0.3 \Leftarrow \|\boldsymbol{x} - \boldsymbol{x}'\| \geq 1.409$
$\quad \delta \leq 0.1 \Leftarrow \|\boldsymbol{x} - \boldsymbol{x}'\| \geq 2.246$
$\quad \delta \leq 0.002 \Leftarrow \|\boldsymbol{x} - \boldsymbol{x}'\| \geq 4.886$

**Matérn-5/2 kernel**
$k(\boldsymbol{x}, \boldsymbol{x}') = \left(1 + \sqrt{5}\|\boldsymbol{x} - \boldsymbol{x}'\| + \frac{5}{3}\|\boldsymbol{x} - \boldsymbol{x}'\|^2\right) \exp\left(-\sqrt{5}\|\boldsymbol{x} - \boldsymbol{x}'\|\right)$:

E.g. $\delta \leq 0.3 \Leftarrow \|\boldsymbol{x} - \boldsymbol{x}'\| \geq 1.457$
$\delta \leq 0.1 \Leftarrow \|\boldsymbol{x} - \boldsymbol{x}'\| \geq 2.214$
$\delta \leq 0.002 \Leftarrow \|\boldsymbol{x} - \boldsymbol{x}'\| \geq 4.485$

## C  GP Local Exploration - Proof

In our main script, we provide a bound of the safety probability. In this section, we provide the proof of this theorem.

We first introduce some necessary theoretical properties in Appendix C.1, and then use the properties to prove Theorem 4.2 and Corollary 4.3 in Appendix C.2.

### C.1  Additional Lemmas

**Definition C.1.** Let $k : \mathcal{X} \times \mathcal{X} \to \mathbb{R}$ be a kernel, $\boldsymbol{A} \subseteq \mathcal{X}$ be any dataset of finite number of elements, and let $\sigma$ be any positive real number, denote $\Omega_{k,\boldsymbol{A},\sigma^2} := k(\boldsymbol{A}, \boldsymbol{A}) + \sigma^2 I$.

**Definition C.2.** Given a kernel $k : \mathcal{X} \times \mathcal{X} \to \mathbb{R}$, dataset $\boldsymbol{A} \subseteq \mathcal{X}$, and some positive real number $\sigma$, then for $\boldsymbol{x} \in \mathcal{X}$, the $k$-, $\boldsymbol{A}$-, and $\sigma^2$-dependent function $\boldsymbol{h}(\boldsymbol{x}) = k(\boldsymbol{A}, \boldsymbol{x})^T \Omega_{k,\boldsymbol{A},\sigma^2}^{-1}$ is called a weight function (Silverman, 1984).

**Proposition C.3.** $C \in \mathbb{R}^{M \times M}$ is a positive definite matrix and $\boldsymbol{b} \in \mathbb{R}^M$ is a vector. $\lambda_{max}$ is the maximum eigenvalue of $C$. We have $\|C\boldsymbol{b}\|_2 \leq \lambda_{max}\|\boldsymbol{b}\|_2$.

*Proof of Proposition C.3.*
Because $C$ is positive definite (symmetric), we can find orthonormal eigenvectors $\{\boldsymbol{e}_1, ..., \boldsymbol{e}_M\}$ of $C$ that form a basis of $\mathbb{R}^M$. Let $\lambda_i$ be the eigenvalue corresponding to $\boldsymbol{e}_i$, we have $\lambda_i > 0$.

As $\{\boldsymbol{e}_1, ..., \boldsymbol{e}_M\}$ is a basis, there exist $b_1, ..., b_M \in \mathbb{R}$ s.t. $\boldsymbol{b} = \sum_{i=1}^M b_i \boldsymbol{e}_i$. Since $\{\boldsymbol{e}_i\}$ is orthonormal, $\|\boldsymbol{b}\|_2^2 = \sum_i b_i^2$. Then

$$\|C\boldsymbol{b}\|_2 = \|\sum_{i=1}^M b_i \lambda_i \boldsymbol{e}_i\|_2 = \sqrt{\sum_{i=1}^M b_i^2 \lambda_i^2}$$

$$\leq \sqrt{\sum_{i=1}^M b_i^2 \lambda_{max}^2} = \lambda_{max}\sqrt{\sum_{i=1}^M b_i^2} = \lambda_{max}\|\boldsymbol{b}\|_2$$

.

$\square$

**Proposition C.4.** $\forall \boldsymbol{A} \subseteq \mathcal{X}$, any kernel $k$, and any positive real number $\sigma$, an eigenvalue $\lambda$ of $\Omega_{k,\boldsymbol{A},\sigma^2}$ (Definition C.1) must satisfy $\lambda \geq \sigma^2$.

*Proof of Proposition C.4.*
Let $\boldsymbol{K} := k(\boldsymbol{A}, \boldsymbol{A})$. We know that

1. $\boldsymbol{K}$ is positive semidefinite, so it has only non-negative eigenvalues, denote the minimal one by $\lambda_K$, and

2. $\sigma^2$ is the only eigenvalue of $\sigma^2 I$.

Then Weyl's inequality immediately gives us the result: $\lambda \geq \lambda_K + \sigma^2 \geq \sigma^2$. $\qquad\square$

**Corollary C.5.** *We are given $\forall \boldsymbol{x}_* \in \mathcal{X}$, $\boldsymbol{A} \subseteq \mathcal{X}$, any kernel $k$ with correlation weakened by distance (Definition 4.1), and any positive real number $\sigma$. Let $M :=$ number of elements of $\boldsymbol{A}$, and let $\boldsymbol{B} \in \mathbb{R}^M$ be a vector. Then $\forall \delta > 0, \exists r > 0$ s.t. when $min_{\boldsymbol{x}' \in \boldsymbol{A}} \|\boldsymbol{x}_* - \boldsymbol{x}'\| \geq r$, we have*

1. *$|\boldsymbol{h}(\boldsymbol{x}_*)\boldsymbol{B}| \leq \sqrt{M}\delta\|\boldsymbol{B}\|/\sigma^2$ (see also Definition C.2),*

2. *$k(\boldsymbol{x}_*, \boldsymbol{x}_*) - k(\boldsymbol{A}, \boldsymbol{x}_*)^T \Omega_{k,\boldsymbol{A},\sigma^2}^{-1} k(\boldsymbol{A}, \boldsymbol{x}_*) \geq k(\boldsymbol{x}_*, \boldsymbol{x}_*) - M\delta^2/\sigma^2$ (see also Definition C.1).*

*Proof of Corollary C.5.*
Let $\boldsymbol{K} := k(\boldsymbol{A}, \boldsymbol{A})$.

Proposition C.4 implies that the eigenvalues of $\left(\boldsymbol{K} + \sigma^2 I\right)^{-1}$ are bounded by $\frac{1}{\sigma^2}$.

In addition, Definition 4.1 gives us $min_{\boldsymbol{x}' \in \boldsymbol{A}}\|\boldsymbol{x}_* - \boldsymbol{x}'\| \geq r \Rightarrow$ all components of row vector $k(\boldsymbol{x}_*, \boldsymbol{A})$ are in region $[0, \delta]$.

1. Apply Cauchy-Schwarz inequality (line 1) and Proposition C.3 (line 2), we obtain

$$
\begin{aligned}
|k(\boldsymbol{A}, \boldsymbol{x}_*)^T \left(k(\boldsymbol{A}, \boldsymbol{A}) + \sigma^2 I\right)^{-1} \boldsymbol{B}| &\leq \|k(\boldsymbol{A}, \boldsymbol{x}_*)^T\|\| \left(\boldsymbol{K} + \sigma^2 I\right)^{-1} \boldsymbol{B}\| \\
&\leq \|k(\boldsymbol{A}, \boldsymbol{x}_*)\|\frac{1}{\sigma^2}\|\boldsymbol{B}\| \\
&\leq \|(\delta, ..., \delta)\|\frac{1}{\sigma^2}\|\boldsymbol{B}\| \\
&\leq \frac{\sqrt{M}\delta\|\boldsymbol{B}\|}{\sigma^2}.
\end{aligned}
$$

2. $\left(\boldsymbol{K} + \sigma^2 I\right)^{-1}$ is positive definite Hermititian matrix, so

$$
\begin{aligned}
k(\boldsymbol{A}, \boldsymbol{x}_*)^T \left(\boldsymbol{K} + \sigma^2 I\right)^{-1} k(\boldsymbol{A}, \boldsymbol{x}_*) &\leq \frac{1}{\sigma^2}\|k(\boldsymbol{A}, \boldsymbol{x}_*)\|^2 \\
&\leq \frac{1}{\sigma^2}M\delta^2.
\end{aligned}
$$

Then, we immediately see that

$$
\begin{aligned}
k(\boldsymbol{x}_*, \boldsymbol{x}_*) - k(\boldsymbol{A}, \boldsymbol{x}_*)^T \left(\boldsymbol{K} + \sigma^2 I\right)^{-1} k(\boldsymbol{A}, \boldsymbol{x}_*) &\geq k(\boldsymbol{x}_*, \boldsymbol{x}_*) - \frac{1}{\sigma^2}\|k(\boldsymbol{A}, \boldsymbol{x}_*)\|^2 \\
&\geq k(\boldsymbol{x}_*, \boldsymbol{x}_*) - \frac{1}{\sigma^2}M\delta^2.
\end{aligned}
$$

$\qquad\square$

**Remark C.6.** $\Phi$ is the cumulative density function (CDF) of a standard Gaussian $\mathcal{N}(0, 1)$. $p(x \leq T) = \Phi(T)$. $p(x \leq -T) = \Phi(-T) = 1 - \Phi(T) = p(x \geq T)$.

## C.2   Main Proof

The theorem is restated again.

**Theorem 4.2** (Local exploration of single-output GPs). *We are given $\boldsymbol{x}_{1:N} \subseteq \mathcal{X}$. For any safety constraint indexed by $j = 1, ..., J$, let $z_{1:N}^j := (z_1^j, ..., z_N^j)$ be the observed noisy safety values and let $\|(z_1^j, ..., z_N^j)\| \leq \sqrt{N}$. The safety value $z^j = q^j(\boldsymbol{x}) + \epsilon_{q^j}$ satisfies the GP assumptions (Assumption 3.1, Assumption 3.2): $q^j \sim \mathcal{GP}(0, k_{q^j}), k_{q^j}(\cdot, \cdot) \leq 1, \epsilon_{q^j} \sim \mathcal{N}\left(0, \sigma_{q^j}^2\right)$. The kernel $k_{q^j}$ is a kernel with correlation weakened by distance (Definition 4.1). Denote $k_{scale}^j := max\ k_{q^j}(\cdot, \cdot)$. Then $\forall \delta \in (0, \sqrt{k_{scale}^j}\sigma_{q^j}/\sqrt{N}), \exists r > 0$ s.t.*

$\forall \boldsymbol{x}_* \in \mathcal{X}$ *that fulfill* $min_{\boldsymbol{x}_i \in \boldsymbol{x}_{1:N}} \|\boldsymbol{x}_* - \boldsymbol{x}_i\| \geq r$, *the probability thresholded on a constant* $T_j$ *is bounded by*
$$p\left((q^j(\boldsymbol{x}_*) \geq T_j)|\boldsymbol{x}_{1:N}, z^j_{1:N}\right) \leq \Phi\left(\frac{N\delta/\sigma^2_{q^j} - T_j}{\sqrt{k^j_{scale} - (\sqrt{N}\delta/\sigma_{q^j})^2}}\right). \quad \Phi \text{ is the CDF of standard Gaussian.}$$

*Proof.*
From Equation (2) in the main script, we know that

$$p\left(q^j(\boldsymbol{x}_*)|\boldsymbol{x}_{1:N}, z^j_{1:N}\right) = \mathcal{N}\left(\boldsymbol{x}_* | \mu_{q^j,N}(\boldsymbol{x}_*), \sigma^2_{q^j,N}(\boldsymbol{x}_*)\right)$$

$$\mu_{q^j,N}(\boldsymbol{x}_*) = k_{q^j}(\boldsymbol{x}_{1:N}, \boldsymbol{x}_*)^T \left(k_{q^j}(\boldsymbol{x}_{1:N}, \boldsymbol{x}_{1:N}) + \sigma^2_{q^j} I_N\right)^{-1} z^j_{1:N}$$

$$\sigma^2_{q^j,N}(\boldsymbol{x}_*) = k_{q^j}(\boldsymbol{x}_*, \boldsymbol{x}_*) - k_{q^j}(\boldsymbol{x}_{1:N}, \boldsymbol{x}_*)^T \left(k_{q^j}(\boldsymbol{x}_{1:N}, \boldsymbol{x}_{1:N}) + \sigma^2_{q^j} I_N\right)^{-1} k_{q^j}(\boldsymbol{x}_{1:N}, \boldsymbol{x}_*).$$

We also know that (Remark C.6)

$$p\left((q^j(\boldsymbol{x}_*) \geq T_j)|\boldsymbol{x}_{1:N}, z^j_{1:N}\right) = 1 - \Phi\left(\frac{T_j - \mu_{q^j,N}(\boldsymbol{x}_*)}{\sigma_{q^j,N}(\boldsymbol{x}_*)}\right)$$

$$= \Phi\left(\frac{\mu_{q^j,N}(\boldsymbol{x}_*) - T_j}{\sigma_{q^j,N}(\boldsymbol{x}_*)}\right).$$

From Corollary C.5, we get $\frac{\mu_{q^j,N}(\boldsymbol{x}_*) - T_j}{\sigma_{q^j,N}(\boldsymbol{x}_*)} \leq \frac{\sqrt{N}\delta \|z^j_{1:N}\|/\sigma^2_{q^j} - T_j}{\sqrt{k_{q^j}(\boldsymbol{x}_*, \boldsymbol{x}_*) - N\delta^2/\sigma^2_{q^j}}}$. This is valid because we assume $\delta < \sqrt{k^j_{scale}}\sigma_{q^j}/\sqrt{N}$. Then with $\|z^j_{1:N}\| \leq \sqrt{N}$ and the fact that $\Phi$ is an increasing function, we immediately see the result

$$p\left((q^j(\boldsymbol{x}_*) \geq T_j)|\boldsymbol{x}_{1:N}, z^j_{1:N}\right) \leq \Phi\left(\frac{N\delta/\sigma^2_{q^j} - T_j}{\sqrt{k^j_{scale} - (\sqrt{N}\delta/\sigma_{q^j})^2}}\right).$$

$\square$

Then, we would like to prove the Corollary 4.3 which is restated here.

**Corollary 4.3** (Existence of $\delta$). *We are given the assumptions in Theorem 4.2. For each* $j = 1, ..., J$, *if either (1)* $T_j \geq 0, \beta^{1/2} > 0$ *or (2)* $T_j < 0, \beta^{1/2} > \frac{|T_j|}{\sqrt{k^j_{scale}}}$, *then* $\exists \delta \in (0, \sqrt{k^j_{scale}}\sigma_{q^j}/\sqrt{N})$ *s.t.*
$\Phi\left(\frac{N\delta/\sigma^2_{q^j} - T_j}{\sqrt{k^j_{scale} - (\sqrt{N}\delta/\sigma_{q^j})^2}}\right) \leq \Phi(\beta^{1/2})$.

*Proof.* This can be proved by substituting the constants.
Condition (1) $T_j \geq 0, \beta^{1/2} > 0$:

$$\frac{N\delta/\sigma^2_{q^j} - T_j}{\sqrt{k^j_{scale} - (\sqrt{N}\delta/\sigma_{q^j})^2}} \leq \frac{N\delta/\sigma^2_{q^j}}{\sqrt{k^j_{scale} - (\sqrt{N}\delta/\sigma_{q^j})^2}},$$

$lim_{\delta \to 0^+} \frac{N\delta/\sigma^2_{q^j}}{\sqrt{k^j_{scale} - (\sqrt{N}\delta/\sigma_{q^j})^2}} = 0$ guarantees $\exists \delta \in (0, \sqrt{k^j_{scale}}\sigma_{q^j}/\sqrt{N})$ s.t. $\frac{N\delta/\sigma^2_{q^j}}{\sqrt{k^j_{scale} - (\sqrt{N}\delta/\sigma_{q^j})^2}} \leq \beta^{1/2}$, for $\beta^{1/2} > 0$. Then because $\Phi$ is strictly increasing, the same $\delta$ gives $\Phi\left(\frac{N\delta/\sigma^2_{q^j} - T_j}{\sqrt{k^j_{scale} - (\sqrt{N}\delta/\sigma_{q^j})^2}}\right) \leq \Phi(\beta^{1/2})$.

Condition (2) $T_j < 0, \beta^{1/2} > \frac{|T_j|}{\sqrt{k^j_{scale}}}$: We see here that $lim_{\delta \to 0^+} \frac{N\delta/\sigma^2_{q^j} - T_j}{\sqrt{k^j_{scale} - (\sqrt{N}\delta/\sigma_{q^j})^2}} = \frac{-T_j}{\sqrt{k^j_{scale}}} < \beta^{1/2}$.
Therefore, there must exist $\delta \in (0, \sqrt{k^j_{scale}}\sigma_{q^j}/\sqrt{N})$ s.t. $\Phi\left(\frac{N\delta/\sigma^2_{q^j} - T_j}{\sqrt{k^j_{scale} - (\sqrt{N}\delta/\sigma_{q^j})^2}}\right) \leq \Phi(\beta^{1/2})$. $\square$

# D  Multi-output GPs with Source Pre-Computation

## D.1  Two-steps Cholesky Decomposition

Given a multi-output GP $g \sim \mathcal{GP}(0, k_g)$, $g \in \{f, q^1, ..., q^J\}$, where $k_g$ is an arbitrary kernel, the main computational challenge is to compute the inverse or Cholesky decomposition of

$$\Omega_g = \begin{pmatrix} K_{g_s} + \sigma_{g_s}^2 I_{N_\text{source}} & K_{g_s,g} \\ K_{g_s,g}^T & K_g + \sigma_g^2 I_N \end{pmatrix}.$$

Such computation has time complexity $\mathcal{O}\left((N_\text{source} + N)^3\right)$. We wish to avoid this computation repeatedly. As in our main script, $k_g$ is parameterized and we write the parameters as $\theta_g = (\theta_{g_s}, \theta_g)$, where $k_g((\cdot, s), (\cdot, s))$ is independent of $\theta_g$.

Here we propose to fix $K_{g_s}$ ($\theta_{g_s}$ must be fixed) and $\sigma_{g_s}^2$ and precompute the Cholesky decomposition of the source components, $L_{g_s} = L(K_{g_s} + \sigma_{g_s}^2 I_{N_\text{source}})$, then

$$L(\Omega_g) = \begin{pmatrix} L_{g_s} & \mathbf{0} \\ \left(L_{g_s}^{-1} K_{g_s,g}\right)^T & L\left(\tilde{K}_t\right) \end{pmatrix},$$

$$\tilde{K}_t = K_g + \sigma_g^2 I_N - \left(L_{g_s}^{-1} K_{g_s,g}\right)^T L_{g_s}^{-1} K_{g_s,g}. \tag{9}$$

This is obtained from the definition of Cholesky decomposition, i.e. $\Omega_g = L(\Omega_g) L(\Omega_g)^T$, and from the fact that a Cholesky decomposition exists and is unique for any positive definite matrix.

The complexity of computing $L(\Omega_g)$ thus becomes $\mathcal{O}(N_\text{source}^2 N) + \mathcal{O}(N_\text{source} N^2) + \mathcal{O}(N^3)$ instead of $\mathcal{O}\left((N_\text{source} + N)^3\right)$. In particular, computing $L_{g_s}^{-1} K_{g,st}$ is $\mathcal{O}(N_\text{source}^2 N)$, acquiring matrix product $\hat{K}_t$ is $\mathcal{O}(N_\text{source} N^2)$ and Cholesky decomposition $L(\hat{K}_t)$ is $\mathcal{O}(N^3)$.

The learning procedure is summarized in Algorithm 3 in the main script. We prepare a safe learning experiment with $\mathcal{D}_{N_\text{source}}^\text{source}$ and initial $\mathcal{D}_N$; we fix $\theta_{f_s}, \theta_{q_s^j}, \sigma_{f_s}, \sigma_{q_s^j}, j = 1, ..., J$ to appropriate values, and we precompute $L_{f_s}, L_{q_s^j}$. During the experiment, the fitting and inference of GPs (for data acquisition) are achieved by incorporating Equation (9) in Equation (5) of the main script (Section 5).

## D.2  Transfer Task GPs beyond One Source Tasks

We extend Section 5.1 beyond one single source task. Let us say we have a total of $P$ source tasks, and the source task index is $s = 1, ..., P$. In our main paper, $\mathcal{D}_{N_\text{source}}^\text{source}$ is the source data with only one task. Here, $\mathcal{D}_{N_\text{source}}^\text{source} := \cup_{s=1}^P \mathcal{D}_{M_s}^s \subseteq \mathcal{X} \times \mathbb{R} \times \mathbb{R}$, $\mathcal{D}_{M_s}^s = \{\boldsymbol{x}_{s,1:M_s}, y_{s,1:M_s}, \boldsymbol{z}_{s,1:M_s}\}$ is the dataset of source task indexed by $s$, $M_s$ is the number of data of task $s$, and $N_\text{source} = \sum_s^P M_s$ is still the number of data of all $P$ source tasks jointly.

We now want to write down the predictive distributions for each $g \in \{f, q^1, ..., q^J\}$. Similar to Section 5.1, $\hat{\boldsymbol{x}}_{s,1:M_s} = \{(\boldsymbol{x}_{s,n}, s)\}_{n=1}^{M_s} \subseteq \mathcal{X} \times \{\text{task indices}\}$ denotes the input data with task index. The data can be plugged in as how it was in Section 5.1, and the predictive distributions have only minor changes. We write $f$ as an example below in Equation (10), while $q^1, ..., q^J$ are analogous. $\hat{\boldsymbol{x}}_* = (\boldsymbol{x}_*, t), \boldsymbol{x}_* \in \mathcal{X}$ is again a test point and $t$ is the index of target task. We color the modification compared to single source task (Equation (5)).

$$p\left(\boldsymbol{f}(\boldsymbol{x}_*, t)|\mathcal{D}_N, \mathcal{D}_{N_{\text{source}}}^{\text{source}}\right) = \mathcal{N}\left(\mu_{\boldsymbol{f},N}(\boldsymbol{x}_*), \sigma_{\boldsymbol{f},N}^2(\boldsymbol{x}_*)\right),$$

$$\mu_{\boldsymbol{f},N}(\boldsymbol{x}_*) = \boldsymbol{v}_f^T \Omega_{\boldsymbol{f}}^{-1} \begin{pmatrix} y_{1,1:M_1} \\ \vdots \\ y_{P,1:M_P} \\ y_{1:N} \end{pmatrix},$$

$$\sigma_{\boldsymbol{f},N}^2(\boldsymbol{x}_*) = k_{\boldsymbol{f}}\left(\hat{\boldsymbol{x}}_*, \hat{\boldsymbol{x}}_*\right) - \boldsymbol{v}_{\boldsymbol{f}}^T \Omega_{\boldsymbol{f}}^{-1} \boldsymbol{v}_{\boldsymbol{f}},$$

$$\boldsymbol{v}_f = \begin{pmatrix} k_{\boldsymbol{f}}(\hat{\boldsymbol{x}}_{1,1:M_1}, \hat{\boldsymbol{x}}_*) \\ \vdots \\ k_{\boldsymbol{f}}(\hat{\boldsymbol{x}}_{P,1:M_P}, \hat{\boldsymbol{x}}_*) \\ k_{\boldsymbol{f}}(\hat{\boldsymbol{x}}_{1:N}, \hat{\boldsymbol{x}}_*) \end{pmatrix}, \tag{10}$$

$$\Omega_{\boldsymbol{f}} = (K_{N_{\text{source}}+N}) + \begin{pmatrix} \sigma_{f_1}^2 I_{M_1} & 0 & & \\ 0 & \ddots & 0 & \\ & 0 & \sigma_{f_P}^2 I_{M_P} & 0 \\ & & 0 & \sigma_f^2 I_N \end{pmatrix},$$

where $[K_{N_{\text{source}}+N}]_{i,j} = k_{\boldsymbol{f}}([\hat{\boldsymbol{x}}_\cup]_i, [\hat{\boldsymbol{x}}_\cup]_j)$, and $\hat{\boldsymbol{x}}_\cup$ is a joint expression of source and target data $(\hat{\boldsymbol{x}}_{s=1,1:M_1}, ..., \hat{\boldsymbol{x}}_{s=P,1:M_P}, \hat{\boldsymbol{x}}_{1:N})$ placed exactly in this order. The GP model $\boldsymbol{f}$ is governed by the multi-task kernel $k_{\boldsymbol{f}}$ and noise parameters $\sigma_{f_s}^2, \sigma_f^2$, where $\sigma_{f_s}^2$ is a noise variance of source task $s = 1, ..., P$. The pre-computation will fix the part of all source tasks (still the top left $N_{\text{source}}$ by $N_{\text{source}}$ block of $\Omega_{\boldsymbol{f}}$).

**Multitask Kernels:** Few examples of actual GP models, i.e. actual kernels, are described as the following. The LMC, linear model of corregionalization, can be taken simply by adding more dimension:

$$k_{\boldsymbol{g}}((\cdot, \cdot), (\cdot, \cdot)) = \sum_{l=1}^{P+1} \left( \boldsymbol{W}_l \boldsymbol{W}_l^T + \begin{pmatrix} \kappa_1 & 0 & & \\ 0 & \ddots & 0 & \\ & 0 & \kappa_P & 0 \\ & & 0 & \kappa \end{pmatrix} \right) \otimes k_l(\cdot, \cdot),$$

where $\boldsymbol{g}$ is a multitask function but does not matter to the expression here, each $k_l : \mathcal{X} \times \mathcal{X} \to \mathbb{R}$ is a standard kernel such as a Matérn-5/2 kernel encoding the $l$-th latent pattern, $\otimes$ is a Kronecker product, and $\boldsymbol{W}_l \in \mathbb{R}^{(P+1)\times 1}$ and $\kappa_1, ..., \kappa_P, \kappa > 0$ are task scale parameters (Álvarez et al., 2012). $l$ is a numbering index used only here.

The HGP can be extended in two ways, models in Poloczek et al. (2017) or in Tighineanu et al. (2022). Here we take the model from Tighineanu et al. (2022):

$$k_{\boldsymbol{g}}((\cdot, \cdot), (\cdot, \cdot)) = \sum_{i=0}^{P} \begin{pmatrix} \boldsymbol{0}^{i\times i} & \boldsymbol{0}^{i\times(P+1-i)} \\ \boldsymbol{0}^{(P+1-i)\times i} & \boldsymbol{1}^{(P+1-i)\times(P+1-i)} \end{pmatrix} \otimes k_i(\cdot, \cdot),$$

where $\boldsymbol{0}^{m\times n}$ and $\boldsymbol{1}^{m\times n}$ are matrices of shape $m$ by $n$ with all elements being zero and one, respectively, $m, n = 0, ..., P$. $k_i(\cdot, \cdot)$ is a standard kernel such as a Matérn-5/2 kernel, $i$ is a numbering index.

# E   Safe Set: Noise-free v.s. Noisy Variables

The safe set can be calculated on noise-free $q^1, ..., q^J$ or noisy variables $z^1, ..., z^J$. The first is useful when the system's criticality depends on the noise-free value and is common in the literature, e.g. Berkenkamp et al. (2020). The second is useful when the criticality depends on the noisy value as e.g. this noisy value triggers an emergency stop. This second scenario of noisy safety values is considered in our main comparison partners work Rothfuss et al. (2022). Therefore, we consider noisy safety values in our experiments.

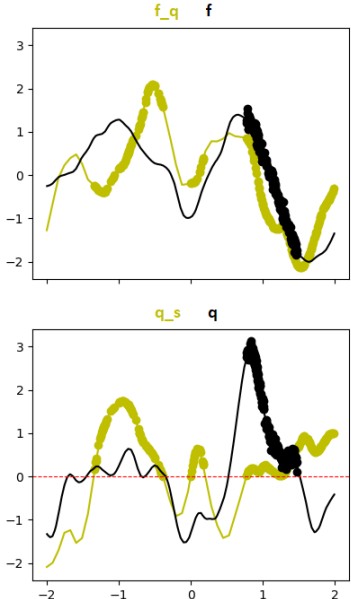

Figure 6: Example simulated GP data of $D = 1$, $\boldsymbol{f}$ is the function we want to learn (top), under an additional safety function $\boldsymbol{q}$ (constraint $\geq 0$, bottom). The curves are true source (yellow) and target (black) functions. The dots are safe source data and a pool of initial target ticket (this pool of target data are more than those actually used in the experiments).

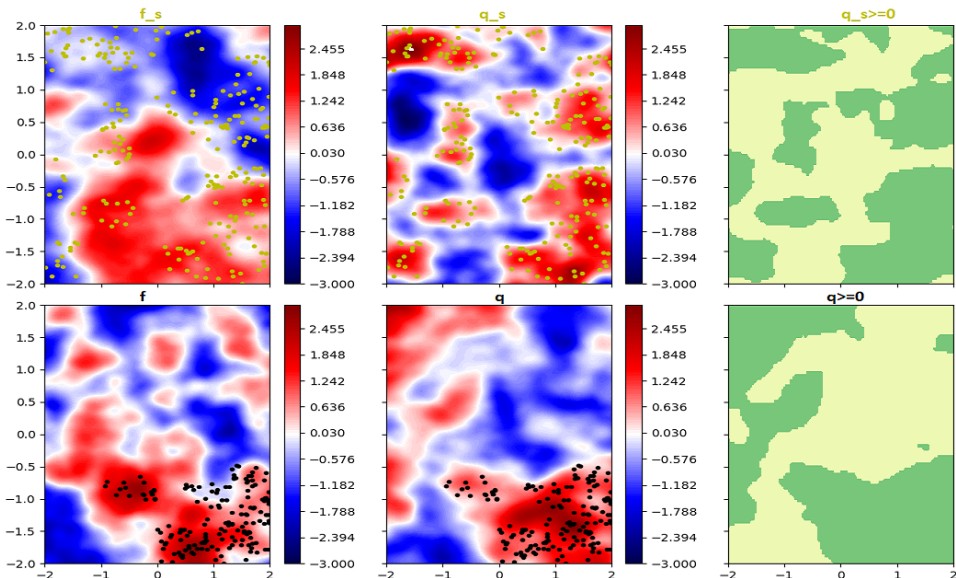

Figure 7: Example simulated GP data of $D = 2$, $\boldsymbol{f}$ is the function we want to learn (left), with an additional safety function $\boldsymbol{q}$ (middle), and the green is true safe regions $\boldsymbol{q} \geq 0$ (right). The top is source task and the bottom is target task. The dots are safe source data and a pool of initial target ticket (this pool of target data are more than those actually used in the experiments).

The result of local exploration in our theoretical analysis are presented on noise-free variables. This leads to the stronger theoretical statement: if noise is added to the safety variables, their uncertainty becomes larger. Therefore, the safe set becomes smaller making exploration even more local.

$\forall j = 1, ..., J$, let us say $z^j(\boldsymbol{x})$ is the predictive noisy value at $\boldsymbol{x}$. We can model with a single-task GP $p\left(z^j(\boldsymbol{x})|q^j(\boldsymbol{x})\right) = \mathcal{N}\left(q^j(\boldsymbol{x}), \sigma_{q^j}^2\right) = \mathcal{N}\left(\mu_{q^j,N}(\boldsymbol{x}), \sigma_{q^j,N}^2(\boldsymbol{x}) + \sigma_{q^j}^2\right)$ or a multitask GP $p\left(z^j(\boldsymbol{x})|\boldsymbol{q}^j(\boldsymbol{x},t)\right) = \mathcal{N}\left(\mu_{\boldsymbol{q}^j,N}(\boldsymbol{x}), \sigma_{\boldsymbol{q}^j,N}^2(\boldsymbol{x}) + \sigma_{q^j}^2\right)$ based on the Gaussian noise assumption (Assumption 3.1) and the fact that Gaussian distributions have additive variances. Consequently, a safe set computed by Algorithms 1 to 3 is $\mathcal{S}_N = \cap_{j=1}^J \{\boldsymbol{x} \in \mathcal{X}_{\text{pool}}|\mu_{q^j,N}(\boldsymbol{x}) - \beta^{1/2}\sqrt{\sigma_{q^j,N}^2(\boldsymbol{x}) + \sigma_{q^j}^2} \geq T_j\}$ (single task, Equation (3)) or $\mathcal{S}_N = \cap_{j=1}^J \{\boldsymbol{x} \in \mathcal{X}_{\text{pool}}|\mu_{\boldsymbol{q}^j,N}(\boldsymbol{x}) - \beta^{1/2}\sqrt{\sigma_{\boldsymbol{q}^j,N}^2(\boldsymbol{x}) + \sigma_{q^j}^2} \geq T_j\}$ (transfer task, Equation (8)). One can see that the uncertainty is larger and the safe set is thus smaller compared to noise-free modeling.

The theoretical result is not affected because $p\left((q^j(\boldsymbol{x}_*) \geq T_j)|\boldsymbol{x}_{1:N}, z_{1:N}^j\right)$, the safety likelihood quantified in our Theorem 4.2, is larger than $p\left((z^j(\boldsymbol{x}_*) \geq T_j)|\boldsymbol{x}_{1:N}, z_{1:N}^j\right)$, indicating that a noisy safe set is bounded by an even smaller value and, therefore, is clearly also bounded by the previously derived quantity.

# F    Experiment & Numerical Details

## F.1    Labeling Safe Regions

The goal is to label disjoint safe regions, so that we may track the exploration of each land. We access safety values as binary labels of equidistant grids (as if these are pixels). This is always possible for synthetic problems. We then perform connected component labeling (CCL, see He et al. (2017)) to the safety classes. This algorithm will cluster safe pixels into connected lands. When $D = 1$, this labeling is trivial. When $D = 2$, we consider 4-neighbors of each pixel (He et al., 2017). For noise-free ground truth safety values, the CCL is deterministic. This algorithm can however be computationally intractable on high dimension (number of grids grows exponentially), and can be inaccurate over real data because the observations are noisy and the grid values need interpolation from the measurements.

After clustering the safe regions over grids, we identify which safe region each test point $\boldsymbol{x}_*$ belongs to by searching the grid nearest to $\boldsymbol{x}_*$. The accuracy can be guaranteed by considering grids denser than the pool. This is computationally possible only for $D = 1, 2$. See main Table 3 and the queried regions count of Figure 11 for the results.

## F.2    Numerical Settup & Datasets

For our main experiments (Algorithm 1, Algorithm 2, Algorithm 3), we set $N_{\text{init}}$ (number of initial observed target data), $N_{\text{source}}$ (number of observed source data), $N_{\text{query}}$ (number of AL queries/ learning iterations) and $N_{\text{pool}}$ (size of discretized input space $\mathcal{X}_{\text{pool}}$) as follows:

1. GP1D: $N_{\text{source}} = 100$, $N_{\text{init}} = 10$, $N_{\text{query}} = 50$, $N_{\text{pool}} = 5000$, constraints $q \geq 0$ up to noise;

2. GP2D: $N_{\text{source}} = 250$, $N_{\text{init}} = 20$, $N_{\text{query}} = 100$, $N_{\text{pool}} = 5000$, constraints $q \geq 0$ up to noise;

3. Branin & Hartmann3: $N_{\text{source}} = 100$, $N_{\text{init}} = 20$, $N_{\text{query}} = 100$, $N_{\text{pool}} = 5000$, $q = f \geq 0$ up to noise;

4. PEngine: $N_{\text{source}} = 500$, $N_{\text{init}} = 20$, $N_{\text{query}} = 100$, and $N_{\text{pool}} = 3000$, constraints $q \leq 1$ up to noise;

5. GEngine: $N_{\text{source}} = 500$, $N_{\text{init}} = 20$, $N_{\text{query}} = 200$, $N_{\text{pool}} = 10000$, $-1.5 \leq q \leq 0.5$ up to noise.

In the following, we describe in details how to prepare each dataset.

### F.2.1    Synthetic Datasets of Tractable Safe Regions

We first sample source and target test functions and then sample initial observations from the functions. With GP1D, GP2D and Branin problems, we reject the sampled functions unless all of the following conditions are satisfied: (i) the target task has at least two disjoint safe regions, (ii) each of these regions has a common safe

area shared with the source, and (iii) for at least two disjoint target safe regions, each aforementioned shared area is larger than 5% of the overall space (in total, at least 10% of the space is safe for both the source and the target tasks).

**GP Data:** We generate datasets of two outputs. The first output is treated as our source task and the second output as the target task.

To generate the multi-output GP datasets, we use GPs with zero mean prior and multi-output kernel $\sum_{l=1}^{2} W_l W_l^T \otimes k_l(\cdot, \cdot)$, where $\otimes$ is the Kronecker product, each $W_l$ is a 2 by 2 matrix and $k_l$ is a unit variance Matérn-5/2 kernel (Álvarez et al., 2012). All components of $W_l$ are generated in the following way: we randomly sample from a uniform distribution over interval $[-1, 1)$, and then the matrix is normalized such that each row of $W_l$ has norm 1. Each $k_l$ has an unit variance and a vector of lengthscale parameters, consisting of $D$ components. For GP1D and GP2D problems, each component of the lengthscale is sampled from a uniform distribution over interval $[0.1, 1)$. We adapt algorithm 1 of Kanagawa et al. (2018) for GP sampling, detailed as follows:

1. sample input dataset $\boldsymbol{X} \in \mathbb{R}^{n \times D}$ within interval $[-2, 2]$, and $n = 100^D$.

2. for $l = 1, 2$, compute Gram matrix $K_l = k_l(\boldsymbol{X}, \boldsymbol{X})$.

3. compute Cholesky decomposition $L_l = L(W_l W_l^T \otimes K_l) = L(W_l W_l^T) \otimes L(K_l)$ (i.e. $W_l W_l^T \otimes K_l = L_l L_l^T$, $L_l \in \mathbb{R}^{2*n \times 2*n}$).

4. for $l = 1, 2$, draw $u_l \sim \mathcal{N}(\boldsymbol{0}, I_{2*n})$ ($u_l \in \mathbb{R}^{(2*n) \times 1}$).

5. obtain noise-free output dataset $\boldsymbol{F} = \sum_{l=1}^{2} L_l u_l$

6. reshape $\boldsymbol{F} = \begin{pmatrix} \boldsymbol{f}(\boldsymbol{X}, s) \\ \boldsymbol{f}(\boldsymbol{X}, t) \end{pmatrix} \in \mathbb{R}^{2*n \times 1}$ into $\boldsymbol{F} = \begin{pmatrix} \boldsymbol{f}(\boldsymbol{X}, s) & \boldsymbol{f}(\boldsymbol{X}, t) \end{pmatrix} \in \mathbb{R}^{n \times 2}$.

7. normalize $\boldsymbol{F}$ again s.t. each column has mean 0 and unit variance.

8. generate initial observations (more than needed in the experiments, always sampled from the largest safe region shared between the source and the target).

During the AL experiments, the generated data $\boldsymbol{X}$ and $\boldsymbol{F}$ are treated as grids. We construct an oracle on continuous space $[-2, 2]^D$ by interpolation. During the experiments, the training data and test data are blurred with a Gaussian noise of standard deviation 0.01 $\mathcal{N}(0, 0.01^2)$.

Once we sample the GP hyperparameters, we sample one main function $\boldsymbol{f}$ and an additional safety function from the GP. During the experiments, the constraint is set to $z_s, z \geq 0$ ($z_s, z$ are noisy $q_s, q$). For each dimension, we generate 10 datasets and repeat the AL experiments 5 times for each dataset. We illustrate examples of $\boldsymbol{X}$ and $\boldsymbol{F}$ in Figure 6 and Figure 7.

**Branin Data:** The Branin function is a function defined over $(x_1, x_2) \in \mathcal{X} = [-5, 10] \times [0, 15]$ as

$$f_{a,b,c,r,s,t}((x_1, x_2)) = a(x_2 - bx_1^2 + cx_1 - r) + s(1-t)cos(x_1) + s,$$

where $a, b, c, r, s, t$ are constants. It is common to set $(a, b, c, r, s, t) = (1, \frac{5.1}{4\pi^2}, \frac{5}{\pi}, 6, 10, \frac{1}{8\pi})$, which is our setting for target task.

We take the numerical setting of Tighineanu et al. (2022); Rothfuss et al. (2022) to generate five different source datasets (and later repeat 5 experiments for each dataset):

$$
\begin{aligned}
a &\sim Uniform(0.5, 1.5), \\
b &\sim Uniform(0.1, 0.15), \\
c &\sim Uniform(1.0, 2.0), \\
r &\sim Uniform(5.0, 7.0), \\
s &\sim Uniform(8.0, 12.0), \\
t &\sim Uniform(0.03, 0.05).
\end{aligned}
$$

After obtaining the constants for our experiments, we sample noise free data points and use the samples to normalize our output

$$
f_{a,b,c,r,s,t}\left((x_1, x_2)\right)_{normalize} = \frac{f_{a,b,c,r,s,t}\left((x_1, x_2)\right) - mean(f_{a,b,c,r,s,t})}{std(f_{a,b,c,r,s,t})}.
$$

Then we set safety constraint $y \geq 0$ ($y$ is noisy $f$) and sample initial safe data. The sampling noise is Gaussian $\mathcal{N}\left(0, 0.01^2\right)$ during the experiments.

### F.2.2 Hartmann3, PEngine, Gengine

**Hartmann3 Data:** Unlike GP and Branin data, we do not enforce disjoint safe regions, and do not track safe regions during the learning. The task generation is not restricted to any safe region characteristics.

The Hartmann3 function is a function defined over $\boldsymbol{x} \in \mathcal{X} = [0, 1]^3$ as

$$
f_{a_1, a_2, a_3, a_4}\left((x_1, x_2, x_3)\right) = -\sum_{i}^{4} a_i exp\left(-\sum_{j=1}^{3} A_{i,j}(x_j - P_{i,j})^2\right),
$$

$$
\boldsymbol{A} = \begin{pmatrix} 3 & 10 & 30 \\ 0.1 & 10 & 35 \\ 3 & 10 & 30 \\ 0.1 & 10 & 35 \end{pmatrix},
$$

$$
\boldsymbol{P} = 10^{-4} \begin{pmatrix} 3689 & 1170 & 2673 \\ 4699 & 4387 & 7470 \\ 1091 & 8732 & 5547 \\ 381 & 5743 & 8828 \end{pmatrix},
$$

where $a_1, a_2, a_3, a_4$ are constants. It is common to set $(a_1, a_2, a_3, a_4) = (1, 1.2, 3, 3.2)$, which is our setting for target task.

We take the numerical setting of Tighineanu et al. (2022) to generate five different source datasets (and later repeat 5 experiments for each dataset):

$$
\begin{aligned}
a_1 &\sim Uniform(1.0, 1.02), \\
a_2 &\sim Uniform(1.18, 1.2), \\
a_3 &\sim Uniform(2.8, 3.0), \\
a_4 &\sim Uniform(3.2, 3.4).
\end{aligned}
$$

After obtaining the constants for our experiments, we sample noise free data points and use the samples to normalize our output

$$
f_{a_1, a_2, a_3, a_4}\left((x_1, x_2, x_3)\right)_{normalize} = \frac{f_{a_1, a_2, a_3, a_4}\left((x_1, x_2, x_3)\right) - mean(f_{a_1, a_2, a_3, a_4})}{std(f_{a_1, a_2, a_3, a_4})}.
$$

Then we set safety constraint $y \geq 0$ ($y$ is noisy $f$) and sample initial safe data. The sampling noise is Gaussian during the experiments $\mathcal{N}\left(0, 0.01^2\right)$.

**PEngine Data:** We have 2 datasets, measured from the same prototype of engine under different conditions. Both datasets measure the temperature, roughness, emission HC, and emission NOx. The inputs are engine speed, relative cylinder air charge, position of camshaft phaser and air-fuel-ratio. The contextual input variables "position of camshaft phaser" and "air-fuel-ratio" are desired to be fixed. These two contextual inputs are recorded with noise, so we interpolate the values with a multi-output GP simulator. We construct a LMC trained with the 2 datasets, each task as one output. During the training, we split each of the datasets (both safe and unsafe) into 60% training data and 40% test data. After the model parameters are selected, the trained models along with full dataset are utilized as our GP simulators (one simulator for each output channel, e.g. temperature simulator, roughness simulator, etc). The first output of each GP simulator is the source task and the second output the target task. The simulators provide GP predictive mean as the observations. During the AL experiments, the input space is a rectangle spanned from the datasets, and $\mathcal{X}_{\text{pool}}$ is a discretization of this space from the simulators with $N_{\text{pool}} = 3000$. We set $N_{\text{source}} = 500$, $N = 20$ (initially) and we query for 100 iterations ($N = 20 + 100$). When we fit the models for simulators, the test RMSEs (60% training and 40% test data) of roughness is around 0.45 and of temperature around 0.25.

In a sequential learning experiment, the surrogate models are trainable GP models. These surrogate models interact with the simulators, i.e. take $\mathcal{X}_{\text{pool}}$ from the simulators, infer the safety and query from $\mathcal{X}_{\text{pool}}$, and then obtain observations from the simulators. In our main Algorithms 1 to 3, the surrogate models are the GP models while the GP simulators are systems that respond to queries $\boldsymbol{x}_*$.

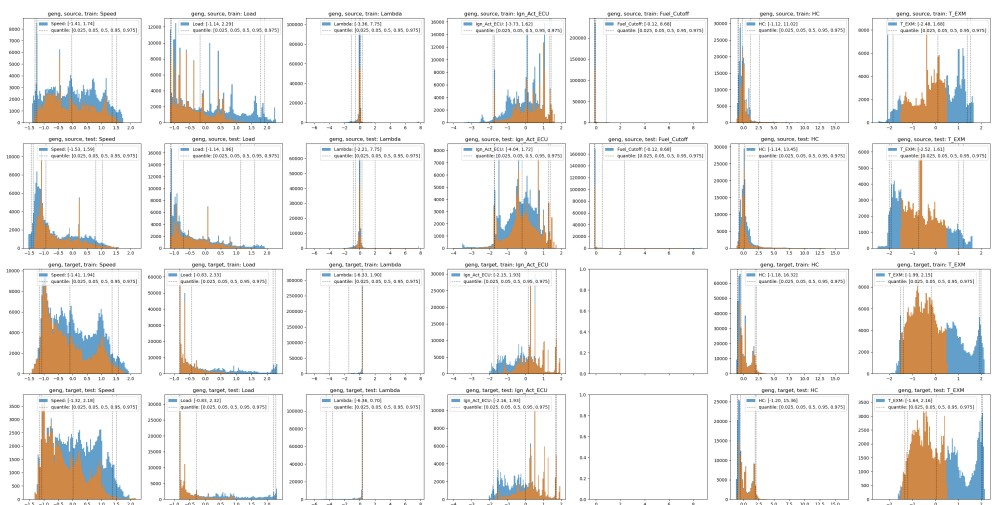

Figure 8: The historgram of GEngine data. The first 5 columns are inputs without NX history structure, the second last column is the output we model with $f, f_s$, and the last column is the temperature constraint variable. The rows are the following in order: (1) source task training set, (2) source task test set (not used in the experiments), (3) target task training set, and (4) target task test set. Blues are the histograms of raw data, and oranges are subsets if we add constraints on the temperature channel.

**GEngine Data:** This problem has two datasets, one taken as the source task and one as the target task. Both datasets were published by Li et al. (2022). Each dataset is split into training set and test set. The original datasets have the following inputs: (1) the first dataset has speed, load, lambda, ignition angle, and fuel cutoff (dimension $D = 5$) which we take as the source task (2) speed, load, lambda, and ignition angle ($D = 4$, no fuel cutoff) which we take as the target task. The 5th input of the source data, fuel cutoff, is irrelevant and we exclude it (not used in the original paper). Please see Figure 8 for the data histogram. The datasets are dynamic and are available with a nonlinear exogenous (NX) history structure, concatenating the

relevant past points into the inputs (handled by Li et al. (2022) in their published code). The final input dimension of this problem is $D = 13$. As outputs, the source dataset measures the temperature, emission particle numbers, CO, CO2, HC, NOx, O2 and temperature. The target dataset measures particle numbers, HC, NOx and temperature. We take HC as our main learning output and temperature as the constraint variable.

Both the source and target datasets have hundreds of thousands of data, but Li et al. (2022) discover that the performance saturates with few thousand randomly selected points or with few hundred actively selected points. We thus decide to run our experiments with $N_{\text{pool}} = 10000$, a random subset of the training set. This pool subset is sampled before we compute the acquisition scores in each iteration. Furthermore, we start our AL experiments with $N_{\text{init}} = 20$ and we query for 200 iterations. The initial target data are sampled from the following input domain (written in the original space, no NX history structure here) $[-1, -0.7] \times (-\infty, -0.5] \times [0, 0.5] \times [0, 0.2]$. This domain is chosen by taking the density peak of the inputs, see row 3 of Figure 8 for the data histogram. Note that values of datasets were normalized.

In this problem, the effect of one single query on the GP hyperparameters is not obvious. Therefore, to speed up the experiments, we train the hyperparameters only every 50 queries (and at the beginning). The constraint is temperature $-1.5 \le z \le 0.5$, and source temperature $-2 \le z_s \le 0.5$. The temperature lower bound matters only to the outliers, it is the upper bound 0.5 that plays the major role. The overall safe set is around 65% of the input space (target test set).

## G   Ablation Studies and Further Experiments

In this section, we provide ablation studies on the size of source dataset.

**One Source Task, Varied $N_{\text{source}}$:**   We perform experiments on the Branin function. The results are presented in Figure 9. The first conclusion is that all of the multitask methods outperform baseline safe AL (safe AL result shown in Figure 4). Note again that the RMSEs are evaluated on the entire space while the baseline safe AL explore only one safe region. In addition, we observe that more source data result in better performances, i.e. lower RMSE and larger safe set coverage (TF area), while there exist a saturation level at around $N_{\text{source}} = 100$.

**Multiple Source Tasks:**   Next, we wish to manipulate the number of source tasks. The transfer task GP formulation and the exact models are described in Appendix D.2. We take LMC and HGP with Matérn-5/2 kernels as the base kernels. In this study, we generate source data with constraints, but discard the disjoint safe regions requirement when we sample the source tasks and data (in Figure 4, the data are generated s.t. source and target task has large enough shared safe area). We consider 1, 3 or 4 source tasks, and we generate 20 or 30 data points per task (Figure 10). In general, we see that 3 source tasks significantly outperform 1 source task while the performance saturates as adding 10 more points per source task seems to benefit more than adding one more source task. Note here that all source data are generated independently, i.e. the observations of each task are not restricted to the same input locations.

**Further Plots and Experiments:**   The main Table 3 and Table 4 present only the summary results. In Figure 11, we additionally provide the region clustering and fitting time w.r.t. AL iterations. Furthermore, Table 5 counts the AL selected queries which, after a safety measurements are accessed, actually satisfy the safety constraints. This table is a sanity check that the methods are selecting points safely.

With the PEngine datasets, we perform additional experiments of learning $\boldsymbol{f} = \boldsymbol{q} =$ temperature, and the results are shown in Figure 12.

Table 5: Ratio of Safe Queries

| Methods | GP1D | GP2D | Branin | Hartmann3 | GEngine |
|---|---|---|---|---|---|
| $N_{\text{query}}$ | 50 | 100 | 100 | 100 | 200 |
| EffTransHGP | $0.986 \pm 0.001$ | $0.974 \pm 0.002$ | $0.999 \pm 0.0006$ | $0.972 \pm 0.003$ | $0.936 \pm 0.003$ |
| FullTransHGP | $0.979 \pm 0.004$ | $0.952 \pm 0.005$ | $0.9996 \pm 0.0004$ | $0.972 \pm 0.003$ | $0.947 \pm 0.01$ |
| FullTransLMC | $0.984 \pm 0.002$ | $0.969 \pm 0.002$ | $0.993 \pm 0.0009$ | $0.968 \pm 0.003$ | $0.91 \pm 0.008$ |
| Rothfuss2022 | $0.975 \pm 0.003$ | $0.905 \pm 0.006$ | $1.0 \pm 0.0$ | $0.84 \pm 0.011$ | $0.765 \pm 0.035$ |
| SAL | $0.995 \pm 0.001$ | $0.958 \pm 0.005$ | $1.0 \pm 0.0$ | $0.966 \pm 0.002$ | $0.954 \pm 0.005$ |

Ratio of all queries selected by the methods which are safe in the ground truth (initial data not included, see Section 6 for the experiments). This is a sanity check in additional to FP safe set area, demonstrates that all the methods are safe during the experiments. Note that our benchmark problems all have around 35% to 65% of the space unsafe. Note that $\beta = 4$ implies that, with a well-fitted safety GP, we tolerate a 2.275% probability of unsafe evaluations. PEngine results are not shown because the queries are all safe (the modeling FP safe set area is almost zero in this problem, see Figure 4 and Figure 12).

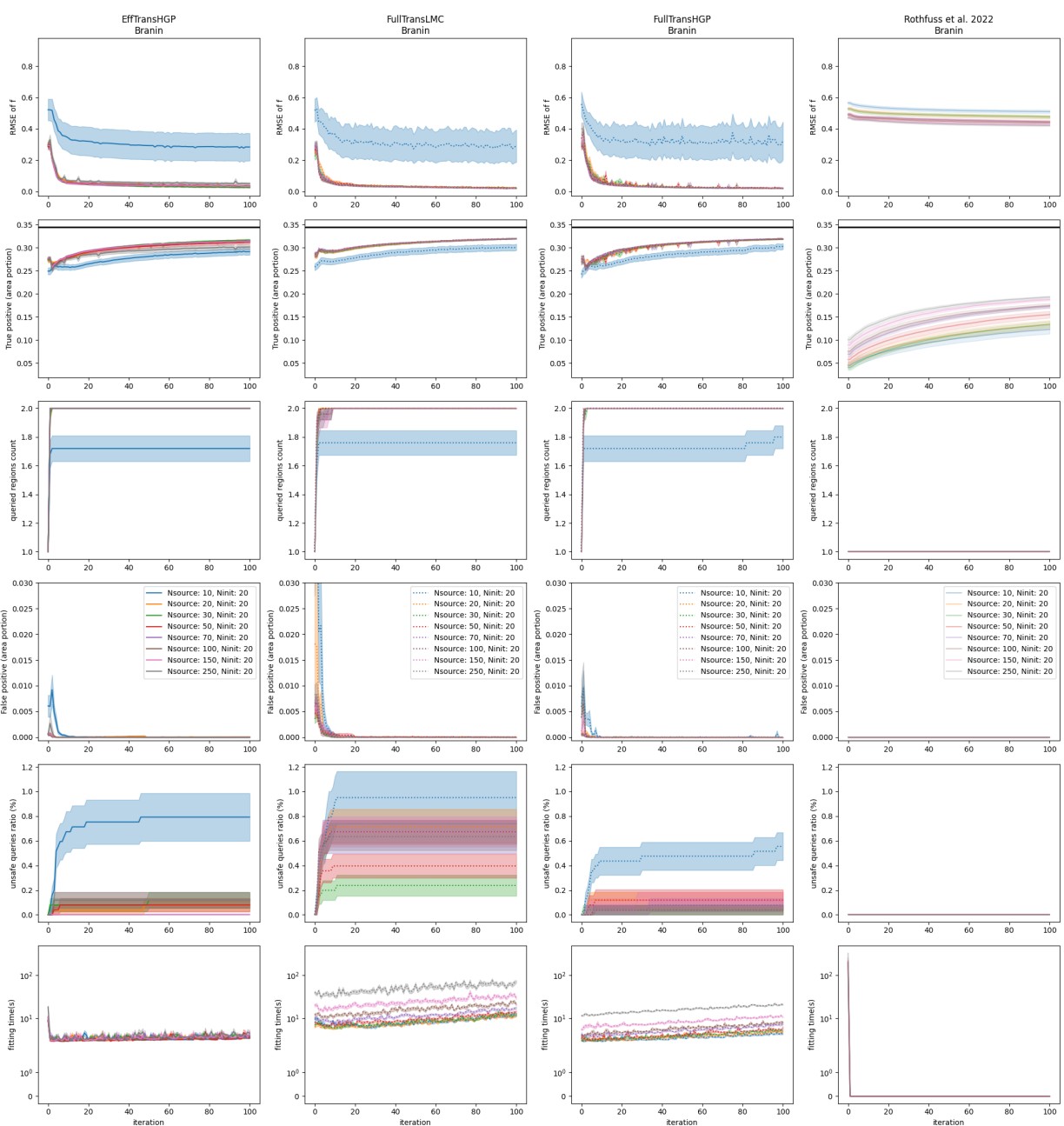

Figure 9: Safe AL experiments: Branin data with different number of source data. Each multitask method is plotted in one column. The results are mean and one standard error of 25 experiments per setting. $\mathcal{X}_{\text{pool}}$ is discretized from $\mathcal{X}$ with $N_{\text{pool}} = 5000$. The TP/FP areas are computed as number of TP/FP points divided by $N_{\text{pool}}$ (i.e. TP/FP as portion of $\mathcal{X}_{\text{pool}}$). The third row shows the number of disjoint safe regions explored by the queries. The fifth row, the unsafe queries ratio, are presented as percentage of number of iterations (e.g. at the 2nd-iteration out of a total of 100 iterations, one of the two queries is unsafe, then the ratio is 1 divided by 100). The last row demonstrates the model fitting time. At the first iteration (iter 0-th), this includes the time for fitting both the source components and the target components (EffTransHGP). With Rothfuss et al. 2022, source fitting is the meta learning phase.

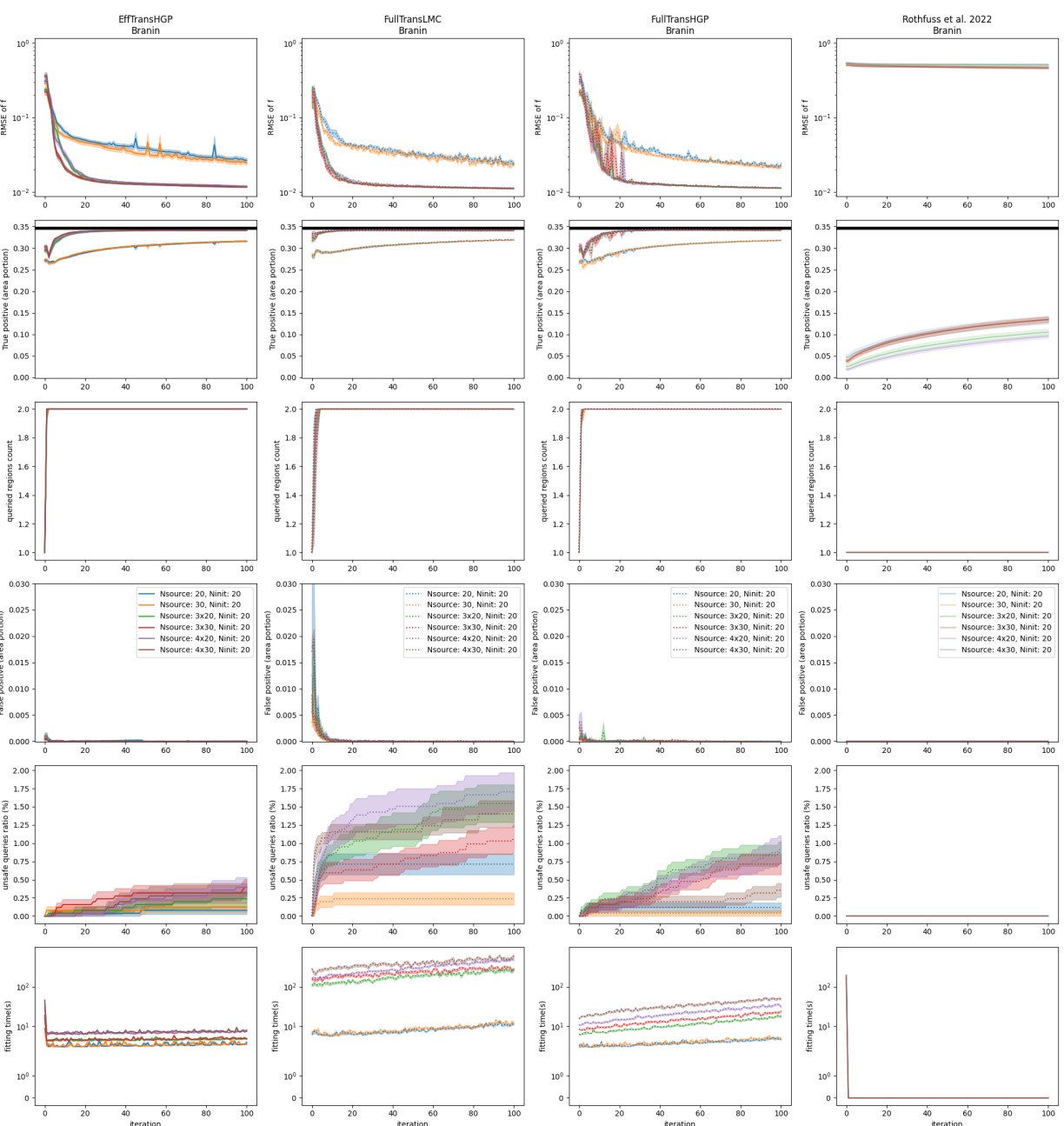

Figure 10: Safe AL experiments with more than one source tasks: Branin data with multiple source tasks. Each multitask method is plotted in one column. We consider 1, 3 or 4 source tasks and sample 20 or 30 data points per task. The remaining setting is the same as described in Figure 9. RMSE plots are plotted in log scale.

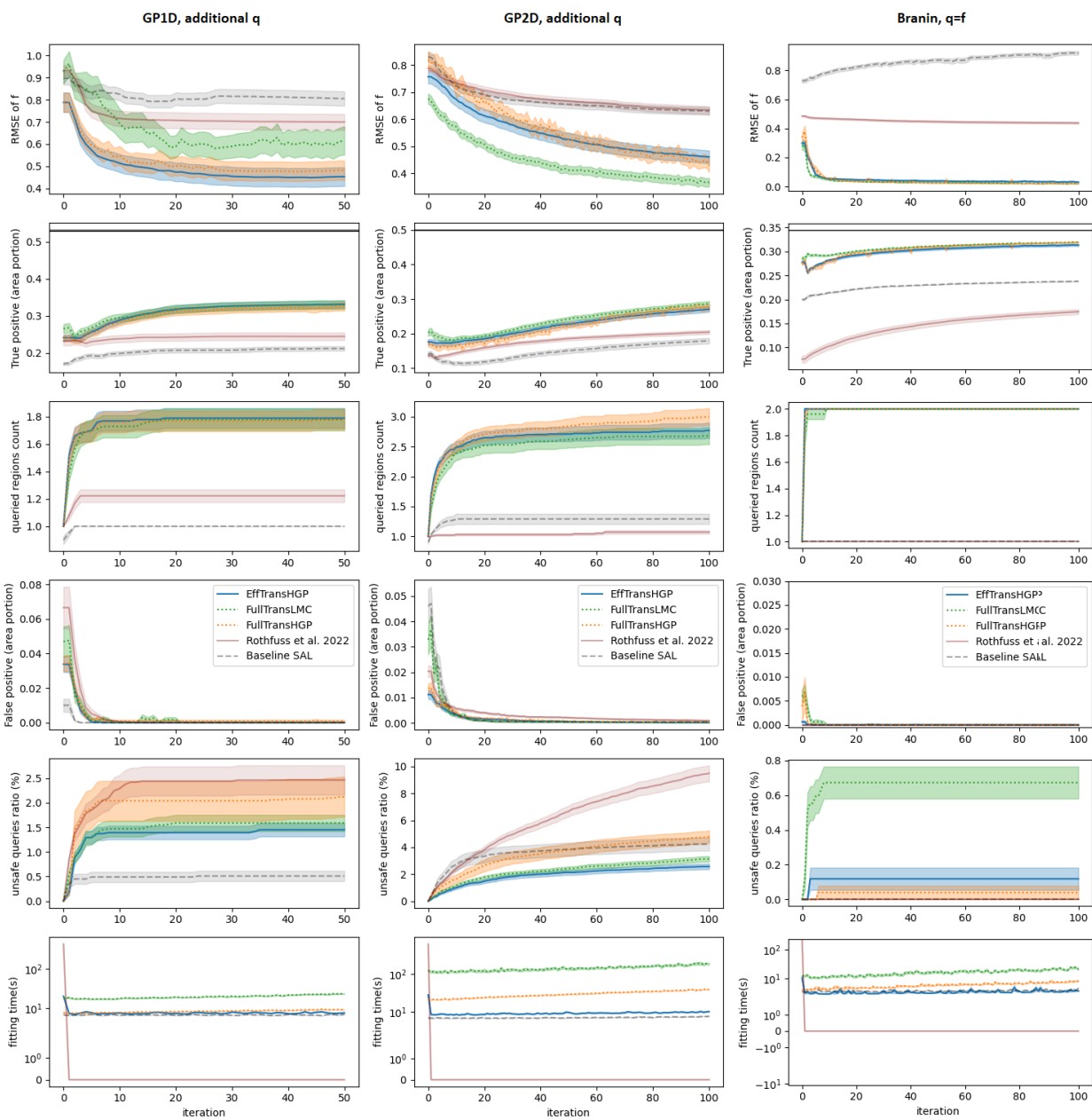

Figure 11: Safe AL experiments on three benchmark datasets: GP data with $\mathcal{X} = [-2, 2]^D$, $D = 1$ or $2$, constrained to $z \geq 0$, and the benchmark Branin function with constraint $y \geq 0$. The results are mean and one standard error of 100 (GP data) or 25 (Branin data) experiments. $\mathcal{X}_{\text{pool}}$ is discretized from $\mathcal{X}$ with $N_{\text{pool}} = 5000$. We set $N_{\text{source}} = 100$ and $N$ is from 10 (0th iteration) to 60 (50th iteration) for GP1D, $N_{\text{source}} = 250$, $N$ is 20 to 120 for GP2D, and $N_{\text{source}} = 100$, $N$ is 20 to 120 for Branin. The first, second and fourth rows are presented in Figure 4 of the main paper. The TP/FP areas are computed as number of TP/FP points divided by $N_{\text{pool}}$ (i.e. TP/FP as portion of $\mathcal{X}_{\text{pool}}$). The third row shows the number of disjoint safe regions explored by the queries (main Table 3 is taken from the last iteration here). The fifth row, the unsafe queries ratio, are presented as percentage of number of iterations (e.g. at the 2nd-iteration out of a total of 50 iterations, one of the two queries is unsafe, then the ratio is 1 divided by 50). The last row demonstrates the model fitting time. At the first iteration (iter 0-th), this includes the time for fitting both the source components and the target components (EffTransHGP). With Rothfuss et al. 2022, source fitting is the meta learning phase.

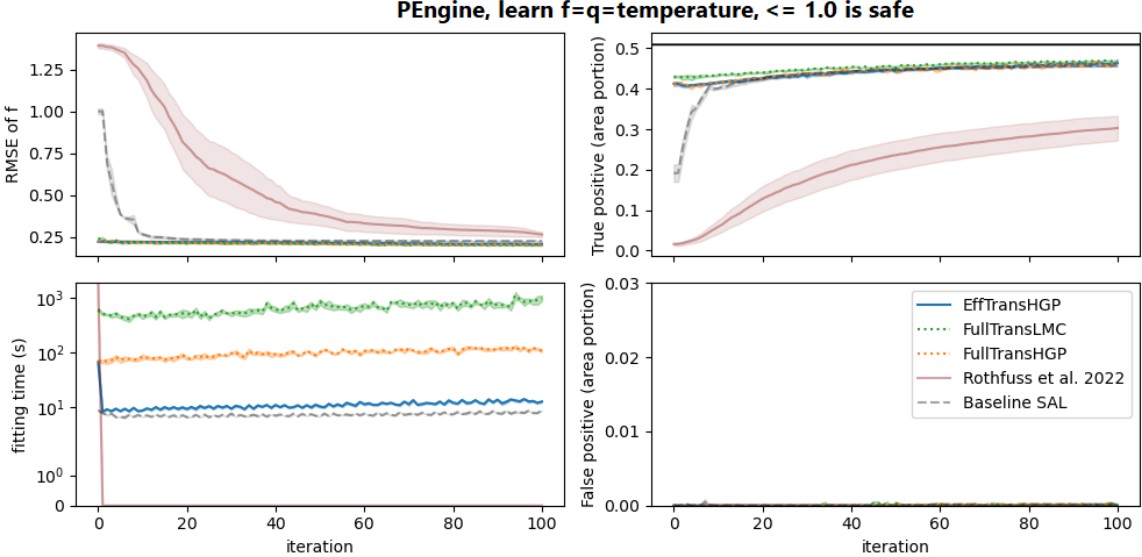

Figure 12: Safe AL experiments on PEngine temperature, AL on $f$ (temperature) constrained by $q = f \leq 1.0$. Baseline is safe AL without source data. Transfer is LMC without modularization. Efficient_transfer is HGP with fixed and pre-computed source knowledge. $N_{\mathrm{source}} = 500$, $N$ is from 20 to 120. The results are mean and one standard error of 5 repetitions. The fitting time is in seconds.

