# OpenReview forum: "Global Safe Sequential Learning via Efficient Knowledge Transfer"
_TMLR — Accepted by TMLR_

### Review · Reviewer_B8BG · 2024-11-07

**Summary Of Contributions:**

This work introduces an interesting approach to safe learning by using transfer learning to enhance the efficiency of safe exploration. Specifically, it addresses the "local exploration" issue encountered by traditional methods (formally illustrated through a theorem) by incorporating data from relevant "source tasks" to support broader "global exploration." However, this added benefit introduces computational challenges, particularly for methods like Gaussian Processes (GPs), where an increased number of data leads to substantial computational overhead. To address this, the authors propose a more efficient kernel computation strategy that fixes blocks related to source data. This approach generalizes the work of Tighineanu et al. (2022), extending the advantages of hierarchical Gaussian Processes (HGP) to a broader class of kernels. Empirical results demonstrate that the proposed approach effectively tackles the local exploration issue, improving both sample efficiency and computation time. This demonstration is conducted within the GP and AL frameworks, assuming that the source and target tasks share similar safety criteria.

**Audience:**

Yes

**Claims And Evidence:**

Yes

**Requested Changes:**

## Writing

I might have overlooked things you already have done. However, I feel the entire problem statement can be organized with a better structure for a smoother presentation. Currently, the elements appear somewhat scattered and placed arbitrarily. The mathematical formulas also need careful attention. Here are some examples, but I highly recommend the authors review the entire article with these points in mind.

1. As a suggestion, you might consider combining “preliminary” with the first paragraph of the “safe learning problem statement” to streamline the protocol description regarding data (e.g., data type, how the data come, etc.). You could then explain how $y$ and $z$ relates to $x$ through functions and the corresponding assumptions. Finally, state the goal regarding evaluating the function $f$ under certain constraints.

2. Again, this is just a suggestion. I don’t think it’s necessary to use so many different formulas to define $z\_{1:N\_{init}}$. This might introduce more confusion than clarity. For example, the following might be sufficient:
The initial safe data $D\_{\text{init}} = \\{ x\_{1:N_{\text{init}}}, y\_{1:N\_{\text{init}}}, z\_{1:N\_{\text{init}}} \\}$ consists of $N\_{\text{init}}$ input samples $x\_{1:N\_{\text{init}}} = \\{ x\_n \\}\_{n=1}^{N\_{\text{init}}} \subseteq X$, corresponding regression outputs $y\_{1:N\_{\text{init}}} = \\{ y\_n \\}\_{n=1}^{N\_{\text{init}}} \subseteq \mathbb{R}$, and safety observations $z\_{1:N\_{\text{init}}} = \\{ z\_n = (z\_n^1, \dots, z\_n^J) \\}\_{n=1}^{N\_{\text{init}}}\subseteq \mathbb{R}^J$.

3. Please remember to specify domains so that quantities are clearly defined. For example, the safety constraints $T_j$ and most of the $\sigma$ terms are not defined, and this also applies to most of the $k_g$ and $\theta$ terms.

4. In several instances, expressions like “$\forall j\in[J]$” and “$\\{q^j\\}_{j=1}^J$” are missing when describing safety-related quantities. Very often, you only cover a specific “$j$”.

5. Adding small words like “let,” “assume,” “where,” and “and” could make the text flow more smoothly. Since there are quite a few notations, it would also help to introduce elements with descriptions as you go, rather than assuming subscripts like “s,” “j,” or “init” will be automatically understood without explanation.

6. Regarding the source data, I noticed that you define $s$ as the index for source tasks, yet in the formulation of $ D\_{N\_{\text{source}}}^{\text{source}} $, all samples seem to belong to a single, specific source task. I understand that in the main paper, you primarily address the case of a single source task and intend to defer the multi-task formulation to the appendix or supplementary material. However, from what I can see, there is no additional formulation provided for multiple source tasks. Consequently, the current formulation in the main paper doesn’t seem to support multiple source tasks.

7. You overload $k_g$ for a lot of different purposes. In Assumption 3.1, it can be function $k_g:X\times X\rightarrow \mathbb{R}$ or a parameter. Later on, it’s used as $k_g:X^N\times X\rightarrow\mathbb{R}^{N\times 1}$, and, without further specification, as $k_g:X^N\times X^N\rightarrow\mathbb{R}^{N\times N}$. Overloading notations is fine, but please ensure it doesn’t create ambiguity.

8. In the statement of Theorem 3.4, you might want to consider placing “for any $x_*\in X$” only before you state $\forall \delta$ …

9. In “Kernel selection”, the notation $\otimes$ and $W_{l,s}$ are not defined.

10. Please ensure consistency in numbering corollaries, theorems, and other items between the appendix and the main paper.

## Ambiguities in explanations

1. In Algorithm 1, is num_step a given quantity? Additionally, it would be helpful to specify in each step what is used as input and what is produced as output. For instance, Lines 2 and 4 in Algorithm 1 are somewhat unclear, and this creates confusion later about why you combine Algorithm 1 with HGP and LMC; as currently stated, Algorithm 1 does not mention source data. I assumed that including source data later on in the experiments implies they are considered part of $D_{N_{\text{init}}}$, which was before only defined to include data from the target task. Hence, please make the data you use more explicitly. Also, in Algorithm 2, a line mentioning the use of decomposition $L$ might add clarity.

2. Regarding the safety constraints, are the unknowns the functions $q_j$ or the thresholds $T_j$ or both? In some cases, such as Example 3.6, the safety constraint function $q$ and the threshold $T = 0$ are clearly defined. However, in examples like GP1D, GP2D, and Branin, the function $q$ and threshold $T$ are not separated, making their interpretation unclear.

3. In your experiments in, e.g., Table 2, how do you define and compute the number of true and explored “regions”?

4. On page 6, could you formally clarify what is meant by “$T_j$ is in the sensitive domain of $g$”? The meaning here is unclear to me.

5. On page 8, could you elaborate on the phrase “cluster the parameters of $k_{f}$ into $\theta_f=(\theta_{f_s}, \theta_f)$”? How exactly is this clustering performed? Also, is this notation essential for the explanation or is it enough to explain merely by the corresponding kernels? Furthermore, why does the clustering not require $k_f$ to be independent of $\theta_{f_s}$? Is this the result of some mathematical properties or does it stem from another consideration?

6. On page 8, what does it mean that the source likelihood “can be barely increased while we explore for the target task”? The formula itself does not appear to change with the target task, so it’s unclear what might drive an increase. Following this, the text states, “Thus, we assume $K_{f_s}$ and $\sigma^2_{f_s}$ remain fixed in the experiments.” Is this reduction in computation based on an assumption or is it a mathematical fact? Either is fine, but it would be useful to clarify.

## Minor

I might have overlooked the description, but are the yellow dots in Figure 1 the source data?

In the second line of page 3, there is some citation/format issue with “Sui et al.”

**Strengths And Weaknesses:**

## Strengths

The work presents a meaningful contribution by combining safe learning with transfer learning, allowing for more efficient and effective safe exploration. Through computationally efficient kernel computation by fixing blocks associated with source data, the authors address a key challenge in, e.g., GPs framework related to computation. This strategy is an extension of Tighineanu et al. (2022) and offers a generalized approach to safe learning that could potentially be applied beyond HGPs. Experimental results suggest that this method does improve exploration efficiency in scenarios where the source and target tasks share similar safety criteria.

## Weaknesses

The paper could benefit from significant improvements in clarity, particularly in the problem statement and sections involving mathematical formulation. When emphasizing the “mathematical formulation” and “theoretical contributions,”you normally don’t want the reader to begin losing confidence in the claim and theoretical soundness when encountering a problem formulation that still falls short of good standards for mathematical rigor.

Although the work provides a theoretical demonstration on the limitations of traditional approaches, it lacks a comparable theoretical validation of the benefits of the proposed method, relying instead on empirical support through experiments.

Additionally, while the authors claim that the method generalizes beyond hierarchical Gaussian Processes (HGP), the paper demonstrates its application using Algorithm 2 only with HGP. This may restrict the perceived impact of the approach, and it remains somewhat unclear to what extent the methods proposed here deviate from those of Tighineanu et al. (2022). Furthermore, as the authors acknowledge in their limitations section, while the work introduces the use of the LMC kernel, this choice does not appear to offer a particularly practical solution.

---

> ### Author Response · Authors · 2024-11-20
>
> We thank the reviewer for the feedback. We address the individual point here.
>
> **Weak. Clarity:**
> We have majorly rewritten the paper to improve clarity. Please see the change summary above.
>
> **Weak. Theory:**
> We have moved appendix figure 6 of the previous version to the main text as new figure 3 to give insights into the benefits of a multitask GP and our new safe AL framework.
>
> **Weak. Modularized on HGP:**
> We acknowledge this limitation, pointed out in section 8. Our modular computation is currently grouped with HGP only, but this scheme provides the possibility for further research on kernel structures.
>
> **Requested Changes. Writing 1-10:**
> We thank the reviewer for the suggestion, and we have revises our paper to improve clarity.
> Please also see our change summary for a revision overview.
> We paid particular attention to the raised points. For example, section 2 is now with only setup and problem statement, and all modeling assumptions are put in GP background section; we tackle the mathematical formulation with more description, and we try to reduce the notation.
>
> **Requested Changes. Ambiguity 1:**
> num\_steps is given, and note that we rename it to $N_{\text{query}}$ which is also listed in table 1.
> We have modified our algorithm 1, 2, 3 to clarify assumptions, quantities that are needed for setup, and we have plugged in exact quantities we compute.
> Note in addition that we separate safe AL and transfer safe AL (no precomputation) into two algorithms with clear distinguishment of modeling assumptions.
>
> **Requested Changes. Ambiguity 2:**
> Safety functions are unknown and need to be learned (the algorithms now specify thresholds).
> For our experiments, the synthetic data always have constraints threshold $0$. To clarify this: (1) we reorganize our experiment section to make all datasets described together like a list, (2) we provide table 2 to summarize all datasets, and (3) we add constraints to the figure title, and we also add constraints more frequently in the appendix.
>
> **Requested Changes. Ambiguity 3:**
> This is done by counting.
> If a region is touched by at least one query, we say this region is discovered.
> This quantification is valid because the safe set can expand from the at least one query.
> The counting is performed with the connected component labeling (CCL) algorithm.
> The CCL algorithm analytically determines disjoint safe lands, which can be indexed for example as land 1, 2, 3.
> Then we look at the safe AL queries to classify whether each query belongs to land 1, 2 or 3.
> This classification has no error on synthetic data GP1D, GP2D, Branin because dense grid points suitable for CCL are available.
> Please see section 7.3 for our revised statement.
>
> **Requested Changes. Ambiguity 4:**
> Yes, thanks for pointing out.
> We have modified the statement there.
> The idea is: if a function $q^{j}$ has values majorly distributed in $[-1, 1]$, then the threshold $T_j$ will also be in $[-1, 1]$, not $T_j = -2$ which is equivalent to an unconstrained problem requiring no safe AL.
> We note additionally that we add corollary 5.3 to clarify the existence of our $\delta$ term, which is used to quantify the explorable bound. Please see section 5 for details.
>
> **Requested Changes. Ambiguity 5:**
> We revise this part and we hope it is much clearer now. $\theta_{f_s}$ are all parameters the source map requires, while all the other parameters are not important for pre-computation and are left as $\theta_f$. This is the idea: the main computational bottleneck is to invert the gram matrix, and it is a big matrix because of the large number of source data. The gram matrix ($(N_{source}+N)\times(N_{source}+N)$) has various subblocks. We aim to fix the source specific block as (1) the data are fixed (offline), and (2) this block takes the main complexity. We thus fix the model parameters needed for this block. In our revision, we reduce the notation to help the clarity.
>
> **Requested Changes. Ambiguity 6:**
> Thank your for pointing out. The original statement was perhaps unclear. The likelihood statement was to describe that the source dataset is large enough, and the learning solely on source saturates. We now remove this to avoid confusion. The second part, fixing the source block, is an “assumption”. We wish to fix the source specific block to save a huge amount of computation. This requires us to fix part of the GP parameters, which however reduces the training flexibility on the target task. This is the tradeoff we need to take for fast decisions in AL iterations. Please see our section 6.2 for a revised version.
>
> **Minor:** yes those are source data and we modify the caption.

---

> > ### Comment · Reviewer_B8BG · 2024-11-27
> >
> > I appreciate the authors' effort in revising the paper. The changes make the content flow smoother and address most concerns I previously raised regarding clarity. I maintain my earlier comments about the strengths and weaknesses of the paper, except for those related to clarity, which has been significantly improved.
> >
> > I have two final comments:
> >
> > 1. In Figure 2, should the black line be referred to as the "safety function" rather than the "safety constraint"? Referring to it as a "constraint" might cause confusion, as constraints could be associated with the T_j-values that define the thresholds to be obeyed.
> >
> > 2. Another suggestion on restructuring sections: You might consider postponing Section 3.2 to a later point in the paper, perhaps around Section 6, and consider the possibility of combining Sections 3.1 and 4. The reasoning is as follows:
> > - Postponing Section 3.2: Sections 3 through 5 (excluding 3.2) are to provide the background on GPs and safe AL, as well as highlighting the limitations of classic approaches, all within the discussion on target tasks. Postponing Section 3.2 (which introduces transfer learning in GPs) to the point where the discussion transitions toward leveraging source data would avoid abrupt shifts in focus.
> > - Combining Sections 3.1 and 4: Section 3.1, although titled "GPs," already includes elements related to safe learning, but the broader context of safe AL is separated into Section 4. Since GPs serve as the foundation of safe AL (as noted in the introduction to Section 3), merging these sections might create a more cohesive introduction.

---

> > > ### Author Response · Authors · 2024-11-28
> > >
> > > Thank you for reviewing the revised manuscript and for the feedback!
> > >
> > > Yes, in the figure, "safety function" is a better term, and we will change it.
> > >
> > > We are also happy to include the proposed restructuring in the next step, if none of the other reviewers objects.

---

> > > > ### Author Response · Authors · 2024-12-05
> > > >
> > > > We have addressed both of the comments and we uploaded the revision.

---

### Review · Reviewer_qhhS · 2024-11-07

**Summary Of Contributions:**

The paper proposes an approach for safe sequential learning by leveraging two key strategies: 1) transfer learning to enhance the exploration of safe regions and improve data efficiency in tasks with potentially multi-modal safety regions, and 2) a modularized approach in multi-output GPs that reduces computational complexity by pre-computing source-relevant components.

The transfer learning aspect allows for the acceleration of learning by utilizing source knowledge to guide the exploration of disjoint safe regions, which is particularly beneficial in scenarios where safety conditions are unknown a priori. The modularized GP approach addresses the computational challenges associated with the cubic time complexity of GPs by fixing certain components related to the source data, thus facilitating more efficient computation during the learning process.

**Audience:**

Yes

**Claims And Evidence:**

Yes

**Requested Changes:**

Please kindly refer to the weaknesses above

**Strengths And Weaknesses:**

### Strengths

1. The paper clearly explains the weakness of conventional safe learning methods in safe exploration, including a theoretical upper bound on the maximum explorable safe regions of conventional safe learning methods.

2. The paper is well-structured and the writing is mostly clear

### Weaknesses
1. As expected, and also mentioned by the authors in the limitations and conclusion, the success of safe exploration by the proposed method largely depends on the positive correlation between safe regions of the source and target problems. When there is no such correlation, how does the method perform, especially, its worst-case performance? Can the authors provide more discussion on whether this existence of good correlation is too strong an assumption to have in real-world settings? Ideally, I think for safety-critical applications, it would be great to have some quantitative lower-bound in terms of prediction error and safety measure (or risk of querying the unsafe region) in relation to the source-target correlation. But I totally understand this is beyond the scope of this paper.

2. Even in the low dimension case, in Figure 3, the proposed methods sometimes have initially higher false positives than the baselines. Would this cause a major issue in safety-critical scenarios, where mistakes, i.e., querying the unsafe regions, are extremely costly so false positives outweigh true positives?

3. It might be that I have not fully understood the problem settings. I wonder, in the safe sequential learning setups, when you mistakenly query an unsafe data point, do you still receive the target label despite the data point being unsafe? If this is allowed, how does relying on a source prior to encourage exploration compare to directly modifying the acquisition function in equation 3? For example, instead of constraining the search space to $S_n$, you give more reward for querying data points with higher entropy under the GPs model $q$ for the safety constraint in addition to its entropy under the predictive function $f$. Will relying on some source prior (which can be good or bad), on average, outperform this heuristic in terms of data efficiency and safety measure?

4. What do the yellow dots represent in Figure 1, top, for "our transfer sequential learning"?

---

> ### Author Response · Authors · 2024-11-20
>
> Thanks for acknowledging our paper, and we discuss the points below.
>
> **Requested Changes. Weak 1:**
> Empirically, we believe this assumption is not too strong. Both positive and negative correlations can be learned. For weak correlation, multi-output GPs may behave similarly to a single task one. We modify our conclusion section to clarify this. Furthermore, we target applications such as simulation to real transfer or serial production where we transfer from one prototype to another. In these tasks, the multitask kernels are typically powerful capturing joint patterns, even when the prior is not perfectly assumed.
>
> **Requested Changes. Weak 2:**
> We use probabilistic safety constraints, so some failures are possible (being absolutely safe on unknown constraints is an open question, and we revise our problem statement to clarify this). Depending on how critical a failure is, we can choose more strict (severe damage to the system) or more modest (some time loss due to emergency shutdown) safety conditions (e.g. tune the beta in eq 8).
> We now like to learn the model (decrease RMSE) as quickly as possible while maintaining safety. This does explicitely mean that we do not aim for zero failures as this would slow down exploration. We only aim for maintaing a safety level and, therefore, as long as we do so, we are allowed to have higher false positives as other methods, e.g. in the initial phase.
> That being said, in figure 3 (new figure 4), the false positive rate of the low dimension cases is below 5 percent which is low. Note again that we consider safety with high probability, not absolutely safe. The appendix table 5 (page 35) also demonstrates the high probability of being safe, actively queried with our methods.
>
> **Requested Changes. Weak 3:**
> There are various problem setups, and we follow Sui et al. 2015 and the follow-up works, where measurements are given even for occasional unsafe queries.
> Discounting the acquisition function was done in Bayesian optimization (e.g. Gelbart et al. 2014, see line 3 of related work where we list more reference). This approach is unsafe, particularly for AL. Note in AL that differential entropy of Gaussian is unbounded $(-\infty, \infty)$. This means, after the algorithm explores the safe regions, the entropy of safe regions become very negative, while the unsafe and unexplored regions still have very positive entropy values. As there is no bound of entropy, it remains challenging to design such an acquisition drop specific to unsafe region. One also needs to take into account that safe regions are also learned and the acquisition drop based on the inference cannot target safe regions (if the entire space has dropped acquisition scores, then it is the same as computing unshifted acquisition scores).
>
> **Requested Changes. Weak 4:**
> Those are source data and we add new description in the caption.

---

### Review · Reviewer_SxvX · 2024-11-08

**Summary Of Contributions:**

The paper introduces a transfer learning method for safe sequential learning with Gaussian processes (GPs). The authors first show that standard GPs with stationary kernels cannot explore regions that are distant from the initial observations. To overcome this, the authors use a multi-output GP which incorporates source data to make the exploration more efficient. Furthermore, the authors propose a modularized approach to multi-output GPs that can reduce the computational burden of the source data. The authors demonstrate the efficacy of their method on a variety of real-world datasets for active learning.

**Audience:**

Yes

**Claims And Evidence:**

No

**Requested Changes:**

My main concern with the paper is the writing. This is not to the standards of TMLR in the current form. I was unable to decipher some of the theoretical results. I would be happy to have a look again and give more feedback on the theory and empirical results and engage in a conversation. The same goes for my "No" response to the claims and evidence, which is based on "clarity" of the arguments presented for the claims, I'll be happy to change my response based on our discussion.

1. The problem statement is not clear mathematically or in text and the assumptions are all over the place. Perhaps have a look at the other accepted papers of TMLR to understand what is the expectation of the journal.
2. The abstract should be rewritten so as to have a better structure but also make clear what the exact problem is that the paper is looking at and the approach that the paper is using. For example, what is being transferred (offline data, which the paper assumes is available in abundance (which is fine but should be reflected in the abstract))? The content is there but needs to be organized better.
3. Equation (4) seems to be important for the transfer learning part. Perhaps explaining it and explaining where exactly transfer learning happens should be made clear.
4. Assumption 3.3 seems logically inconsistent is the order correct? Then there exists a variable that is the one implying the condition? Perhaps you meant the other way?
5. Add references for the statement "These learning algorithms often utilize Gaussian processes “ with respect to Active learning. Also explain in the paper why safety is a critical point in active learning. I don’t understand why safety constraints would be there while annotating something? Perhaps an example?
6. Since Figure 1 is the main motivating figure of the paper it should have Descriptive Caption. Also Figure 2 and Figure 3 should have a take-home mes
7. Please add what is the implication of (4) and where exactly is the transfer learning part?
8. What is D, J in preliminary?
9. What is index_space in section 4?
10. Why do you have t=0 and s=1,..? This is a really confusing notation. Just say t is the index of the target tasks and s the notation of the source tasks. If you are considering only a single source and target task in the main paper, you can just use s and t, and then have a subsection or footnote explain their extension and point to the appendix
11. In Notations, Why is z^j_1:N a set? Isn’t this ordered information?
12. What is the condition on the function f, g? It is mentioned first in Assumption 2.1 but much later made clear the assumption 3.1
13. Phi is not defined in the main text to the best of my reading, it is the CDF of the normal function (which I inferred from the proof)
14. g = f, q1 , … \in (instead of =)
15. Please label the algorithms appropriately. Sequential learning is a very broad term that can be used for MDPs, Bandits, and even deterministic optimization.
16. For each theoretical and numerical result central to your argument there should be a) motivation b) intuition and c) implication.
Also, one suggestion to the author is to ensure that each important section of the paper is written so that if a researcher familiar with the field wants to understand and use your results, they can do so without much effort, i.e., the text is accessible. This can greatly help the readability of the paper and also help your work garner more attention.

**Strengths And Weaknesses:**

Strengths:
- Deals with an interesting problem setup: The paper addresses the problem of safe sequential learning with Gaussian processes, which is a relevant and challenging area of research.
- Attempts at reducing computational complexity of the problem: The authors introduce a modularized approach to multi-output GPs that reduces the computational complexity of incorporating source data. This is important in real-world applications where large amounts of source data may be available.
- Has sufficient experiments for a theory paper: The authors conduct experiments on a variety of real-world datasets to demonstrate the effectiveness of their proposed method. The experiments cover different scenarios, including cases with multiple disjoint safe regions and high-dimensional data.

Weakness:

- Poorly Written & Lacks Clarity: The paper's clarity and organization could be significantly improved.
  1) “we introduce the idea of transfer safe sequential learning supported by a thorough mathematical formulation” : From a mathematical standpoint the paper has sloppy writing. I mention several of such mistakes in the requested changes.

- Experiment don't have a message: While the paper includes several experiments, the presentation of the results and the takeaways could be more concise and impactful.
   1. There are 3 claims that the authors make that the experiments should verify, however in the experiment section it is not clear what is the performance improvement with respect to these claims and how much are they verified
   2. It is unclear as to what is the safety aspect in the experiments from a practical standpoint. In general, what is the safety constraint parallel in active learning?


- Intuition for the theory is absent: Although the paper presents theoretical analysis and proofs, it sometimes lacks the necessary intuition to help readers understand the underlying concepts
   1. Theorem 3.4 can lead to a vacous bound if the sigma_f is too big,
   2. There should be some intuition to the proofs in the main paper.
   3. The problem statement is not clear mathematically in the text.
   4. Also what is the theoretical improvement of the results you propose

---

> ### Author Response · Authors · 2024-11-20
>
> We thank the reviewer for the positive feedback on the impact of our research problem and the experiments.
> We discuss the comments and requested changes below.
>
> **Weak 1. Clarity:**
> We have revised our paper, re-structured the sections, and made mathematical formulation clearer.
> Please also see the above change summary for an overview.
>
> **Weak 2.1 Experiment presentation:**
> In our paper, our main objectives are (1) safe set coverage and (2) learning result over the safe set.
> Therefore, either of the following is good: identify larger safe region with fewer queries, or get higher modeling performance with fewer queries.
> To further clarify this, we have (1) revised the problem statement to make clearer description that we aim for model learning on larger safe set, (2) re-structured the experiment sections and re-vised figure captions.
>
> **Weak 2.2, Requested Changes 5. Example of safe AL:**
> We added citations, and we added an example in the introduction.
> We also list few examples here. In engineering, an example is robotics. One wishes to evaluate the performance of robotic controllers. The annotation includes execution of each controller with selected parameters. However, bad parameters can result in unsafe behavior, for example a drone flies in high speed towards a human. This means we have a constraint on the output space (e.g. speed of drone), while data are selected on the input space and the constrained domain is unclear. Another example is compound evaluations. Chemical compounds are often actively chosen because it is unrealistic to synthesize and evaluate all compounds/proteins (annotation includes compound syntheses and experimental tests). This is safety critical because some compounds are dangerous, e.g. in an extreme case they might explode during synthesis.
>
> **Weak 3.1:**
> Vacuous bound is ruled out due to the range of $\delta$, which we highlight in the paper now (section 5).
> We additionally add corollary 5.3 to discuss the existence of our $\delta$ term. The core idea is, for common selection of the safety level $\beta$, our main theorem is valid for the quantification of explorable bound. Please see section 5 for the revised version.
>
> **Weak 3.2:**
> We add more description to improve the clarity.
>
> **Weak 3.3, Requested Changes 1. Clarity of problem statement \& assumptions:**
> We have revised the problem statement to explain safe AL, including what safe queries mean, what we aim to learn, and what we want to accomplish with transfer learning. We have also moved all assumptions to GP background section (section 3).
>
> **Weak 3.4:**
> We leverage transfer learning to overcome the local exploration.
> We have moved an appendix figure to main figure 3 to strengthen this point (section 5).
>
> **Requested Changes 1:** See above **Weak 3.3, Requested Changes 1**.
>
> **Requested Changes 2. Abstract:** We have revised the abstract to address this.
>
> **Requested Changes 3, 7. What is transferred:**
> Transfer happens in the predictive distributions, which are used to compute safe set and acquisition functions. We elaborate in section 6 and we update algorithm 2, 3 to incorporate explicitly where source data are used.
>
> **Requested Changes 4:**
> This assumption described a kernel with a relation between covariance and data distance. We have now made this clearer by removing it and introducing a definition instead. It states that distant points are weakly correlated. This is a property of the commonly used kernels (RBF and Materns all have this property). It is this property that helps us derive the given explorable radius. We add more description into the text (section 5). Please also see our main theorem for the updated description. We also re-organize the sentence to improve clarity
>
> **Requested Changes 5:** See above **Weak 2.2, Requested Change 5**.
>
> **Requested Changes 6, 8:**
> We add figure captions with take home messages; we describe $D$, input dimension, and $J$, number of constraints, in section 2.
>
> **Requested Changes 9:** This is the space of task indices ($\{s, t\}$), but we replace it now by $\{\text{task indices}\}$ and we reduce the frequency of using this term.
>
> **Requested Changes 10-11, 13-15:**
> We further reduce the notation whenever possible and add more description to each notation. $z^{j}_{1:N}$ in the theorem is ordered, and we modify the statement to clarify (theorem 5.2). We modify the algorithms.
>
> **Requested Changes 12:**
> Conditions of functions depend on the modeling scheme. GPs and multitask GPs take different modeling assumptions. We now introduce the modeling assumptions collectively in background GP section (section 3), and we specify the exact underlying assumptions of each algorithm (alg 1-3).
>
> **Requested Changes 16:**
> We address this by restructuring our paper, please see the change summary above. In particular, we segment our sections more often and give each an informative title.

---

> > ### Comment · Reviewer_SxvX · 2024-11-22
> >
> > The authors seem to have done most of the changes I requested. Let me go through the paper once more and see if I have any clarification questions.

---

### Author Response · Authors · 2024-11-20
**shared feedback**

We thank all the reviewers for the effort and the constructive feedback. Our paper “addresses the problem of safe sequential learning with Gaussian processes, which is a relevant and challenging area of research” (Rev. SxvX). We explain “the weakness of conventional safe learning methods” (Rev. qhhS) by deriving a “theoretical upper bound on the maximum explorable safe regions” (Rev. qhhS). We “presents a meaningful contribution by combining safe learning with transfer learning” (Rev. B8BG), and we further “introduce a modularized approach to multi-output GPs that reduces the computational complexity of incorporating source data” (Rev. SxvS), which “is important in real-world applications where large amounts of source data may be available” (Rev. SxvS). We also provide empirical evidence of our algorithms, supported by experiments of “different scenarios, including cases with multiple disjoint safe regions and high-dimensional data” (Rev. SxvS).

The main concerns of the reviewers were related to structure and clarity. In our revision, we majorly re-structured and re-wrote the text to improve the clarity. The modification is summarized in the change summary above. Then, we address each point individually below.

We will be happy to engage in further discussion.

---

### Author Response · Authors · 2025-01-10

Dear AE and reviewers,

Thank you to all for the constructive feedback during the rebuttal period.

We have uploaded the camera ready version. In addition, we provide the link to the code of our paper.

---

### Decision · Action_Editor_UUV1 · 2024-12-09

**Recommendation:** Accept as is

**Comment:**

This paper proposes a new approach to safe active learning in Bayesian optimization. The authors demonstrate the limitations of classic approaches, and validate their approach on both synthetic and real-world problems. The initial reviews of this paper were positive and mostly raised clarity concerns:

* Cumbersome writing

* Unclear notation

* No intuition in theory

* Poorly justified experiments

The authors completely revised the paper and took the many comments of the reviewers into account. The reviewers recognized this effort and unanimously agreed that the paper should be accepted.

**Audience:**

This paper is on the intersection of Bayesian optimization, active learning, and safe learning. The intersection of the disciplines is interesting, and will allow each community to learn from the others.

**Claims And Evidence:**

Yes. This paper proposes a sound approach to safe Bayesian optimization. It demonstrates the limitations of classic approaches to safe active learning, and validates the proposed approach on both synthetic and real-world problems.